



# Dynamics of Gap Winds in the Great Rift Valley, Ethiopia: Emphasis on Strong Winds at Lake Abaya

Cornelius Immanuel Weiß[1], Alexander Gohm[1], Mathias Walter Rotach[1], and Thomas Torora Minda[2]

[1]Department of Atmospheric and Cryospheric Sciences (ACINN), University of Innsbruck, Innrain 52f, 6020 Innsbruck, Austria

[2]Faculty of Meteorology and Hydrology, Arba Minch Water Technology Institute, Arba Minch University, P.O.Box 2221, Arba Minch, Ethiopia

**Correspondence:** Cornelius Immanuel Weiß (weiss-cornelius@gmx.de)

**Abstract.** Lake Abaya, located in the Great Rift Valley (GRV) in Ethiopia, is affected by regularly occurring strong winds that cause water waves, which in turn affect the lake's ecology and food web. The driving forces for these winds, however, are yet unexplained. Hence, the main goal of this study was to provide a physical explanation for the formation of these strong winds in the GRV and especially at Lake Abaya. Therefore, two case studies were performed based on measurements, ERA5 reanalysis data and mesoscale numerical simulations conducted with the Weather Research and Forecasting (WRF) model. The simulations revealed that in both cases a gap flow downstream of the narrowest and highest part of the GRV (i.e. the pass) led to high wind speeds of up to $25\,\mathrm{m\,s^{-1}}$. Two types of gap flow were identified: a northeast gap flow and a southwest gap flow. The wind directions are in line with the orientation of the valley axis and depend on the air mass distribution north and south of the valley and the resulting along-valley pressure gradient. The air mass distribution was determined by the position of the Intertropical Convergence Zone relative to the GRV. The colder air mass was upstream of the GRV in both case studies. During the day, the convective boundary layer in the warmer air mass on the downstream side heated up stronger and quicker than in the colder air mass. The most suitable variable describing the timing of the gap flow was found to be the pressure gradient at pass height, which corresponds roughly to the $800\,\mathrm{hPa}$ pressure level. In both cases the gap flow exhibited a strong daily cycle, which illustrates the importance of the thermal forcing due to differential heating over complex terrain in addition to the large-scale forcing due to air mass differences. The start, strength and the duration of the gap winds within in the valley were location-dependent. For both cases, the strongest winds occurred after sunset and in the ongoing night downstream of the gap and on the corresponding lee slope. The ERA5 reanalysis captures both events qualitatively well but with weaker wind speeds than in the mesoscale numerical simulations. Hence, ERA5 is suitable for a future climatological analysis of these gap flows.

## 1 Introduction

Local scientists [1] observed several cases of comparatively strong winds at Lake Abaya that produce water waves and, hence, affect the lake's ecology. Different wind directions were observed among different cases. According to local fishermen, *Galana Winds* blow from the east, *Dega Winds* from the west, *Gidabo Winds* (also *Sidamo Winds* from the northeast and *Chamo Winds*

---

[1]Thomas Minda (Meteorologist) and Fassil Teffera (Aquatic Ecologist), both from Arba Minch University





from the southwest, respectively. The first two wind directions are perpendicular, the latter two are in line with the valley orientation of the Great Rift Valley (GRV) in Ethiopia (also referred to as Ethiopian Rift Valley or Eastern Rift Valley). *Gidabo*
*Winds* were reported to be the strongest and to occur during nighttime and in the early morning. The physical mechanisms for these strong winds on Lake Abaya, however, are unknown so far.

Lake Abaya is located in the southwest of Ethiopia (6°5'N, 37°41'E) and is the largest of eight lakes in the GRV (Fig. 1a–c). The lake surfaces is situated at an altitude of 1175 m above mean sea level (a.m.s.l.) and is roughly 70 km long and 5 to 25 km wide. The average depth is 8.6 m (Teffera et al., 2017). Its elongated shape is in line with the valley orientation from northeast to
southwest. East and west of the lake, topography rises to more than 3000 m a.m.s.l. within 30 km. From the northern shoreline, the valley floor rises monotonously towards a pass at around 2000 m a.m.s.l. near the city of Hawassa (close to PA in Fig. 1b). North of the pass the height of the valley floor remains unchanged for roughly 150 km before it declines and widens into the Afar Triangle in northeastern Ethiopia. Approximately 5 km south of Lake Abaya follows Lake Chamo (Fig. 1c). These two lakes are separated by a shallow ridge called Yegzier Dildiyi (meaning *God's Bridge*) with an elevation of 400 m above ground
level (a.g.l.). The southern end of the GRV in Ethiopia is determined by the Turkana Channel at the border to Kenya.

Lake Abaya plays a key role in the local socio-ecological system and has a high biological diversity (Zinabu et al., 2002). The lake serves as fishery as well as water supply for the local human population, wildlife and agriculture (e.g., banana cultivation). Four decades ago, the ecology started to change. Due to massive deforestation and fertilization in the catchment of Lake Abaya, an immense input of sediments and nutrient runoff is present (Teffera, 2016; Zekarias et al., 2021). This causes not
only a severe eutrophication of the lake but also a high turbidity (Teffera et al., 2017; Lemmens et al., 2017). Higher turbidity impedes insolation that is necessary for primary production of phyto- and zooplankton (Lemmens et al., 2017). Hence, the food web of Lake Abaya is threatened and has to adapt. Furthermore, Lake Abaya as a fresh water lake is subject to the growth of water hyacinth (Mengistu et al., 2017). This buoyant plant covers the lake surface. Therefore, the weak solar penetration into the lake is even further mitigated. Due to its large surface, Lake Abaya is prone to generate waves as a result of wind stress.
This has a bidirectional effect on the lake ecology. On the one hand, wind-induced sediment resuspension keeps turbidity near the surface high (Scheffer, 2004; Gebremariam, 2009). On the other hand, strong waves have the ability to destroy the colonies of water hyacinth and other plants on the lake surface and thereby increase the penetration of shortwave radiation. Disrupted water hyacinths have been observed and linked to the occurrence of strong winds and waves (T. Minda and F. Teffera, personal communication). The waves could be the reason why Lake Abaya is less affected by invasive plants than other Ethiopian lakes
(e.g., Lake Tana).

The synoptic wind field in tropical areas is generally influenced by the displacement of the Intertropical Convergence Zone (ITCZ) (Korecha and Sorteberg, 2013; Gleixner et al., 2017). Besides other large scale drivers like ENSO, this low pressure belt is the key synoptic feature which determines the weather in Ethiopia (Minda, 2019). The ITCZ reaches its northernmost position over Africa at around 15°N in June and its extreme southward position at 15°S in January (Diro et al., 2007; Sjoukje
et al., 2018). Hence, the general direction of the surface winds is determined by the associated trade winds underneath the trade wind inversion which is at around 2000 m a.m.s.l. according to Cao et al. (2007) and Carrillo et al. (2016). Therefore, the prevailing wind direction is from the northeast around January and from the southeast around June (Minda et al., 2018a). Due





**Figure 1.** (a) Topographic map of East Africa with the two nested WRF model domains d01 and d02 indicated by the red boxes. The area contains the Afar Triangle, the Turkana Channel and the Great Rift Valley (GRV) in between. (b) Topographic map of d02 and the GRV. Along-valley cross sections marked in blue (a-b-c, a′-b and b′-c′). Several grid points of the WRF model (red circles) and the ERA5 reanalysis (dark red crosses) used in the investigations are indicated by markers and two-letter labels (see legend). (c) Topographic map of the target area around Lake Abaya and Lake Chamo. This subdomain is also indicated by a black dashed box in (b). Blue stars indicate the location of automatic weather stations of the Gamo Ethiopian Meteorological Stations (GEMS) network. Yellow circles mark additional landmarks.

to the dual passage of the ITCZ, the synoptic wind direction in Ethiopia oscillates between south and north by turning over east.





The position of the ITCZ is not the only factor that determines the wind field in East Africa. Ethiopia is a country rich
in lakes and complex topography. Hence, lake- and terrain-induced mesoscale winds may dominate the local flow near the
surface, as found by Haile et al. (2009) for the Lake Tana basin, the largest lake in Ethiopia. The adjustment of the wind field
by local effects strongly varies on the daily and seasonal scale (Minda et al., 2018a).

For the particular region of interest at Lake Abaya, detailed studies concerning the local wind field are rare. Gebremariam
(2009) found the observed lake and land breeze in February (the beginning of *belg* season) to be the dominant wind feature
which shows strong differences to the synoptic flow (ITCZ south of the equator). Minda (2019) stated that the temperature
contrast between day and night during this time of the year is large, supporting the development of mesoscale thermally driven
flows like the lake and land breeze but also valley winds. Nevertheless, Gebremariam (2009) found that the wind field around
Lake Abaya is at least in some months connected to the large scale flow. According to Gebremariam (2009) the connection is
strongest in May and June (end of *belg* and beginning of *kirmet* season), when the temperature contrast between day and night
is weakest. In addition to the lake and land breeze, Minda et al. (2018a) observed anabatic and katabatic winds at the slopes of
the valley.

In conclusion, strong winds at Lake Abaya have a major impact on the lake water dynamics, hence, on food production and
thus play an important role in the everyday life of the local community. Therefore, these winds need to be better understood
for eco-meteorological benefits. Since, to the best of our knowledge, there is not a single study dealing with the topic of
strong wind events in the GRV, the goal of this study is to clarify the physical mechanisms of two cases representative for two
different seasons. A special focus will be on the questions whether the phenomenon is dynamically or thermally driven and
if a daytime or seasonal dependency exists. The paper is organized as follows. Section 2 presents the data and methods used
for the investigation. The synoptic conditions prior to the two cases are shortly summarized in Sect. 3, followed by a detailed
description of the two events in Sect. 4 and Sect.5. The results are discussed in Sect. 6 and conclusions are drawn in Sect. 7.

## 2   Data and methods

### 2.1   Observations

Observations from the Gamo Ethiopian Meteorological Stations (GEMS) network (Minda et al., 2018b), a local network of
automatic weather stations (AWS) in the area around Lake Abaya, were used to identify strong wind events at Lake Abaya.
The locations of the considered AWS are marked in Fig. 1c. The detailed procedure of the analysis, as well as a list of 20 cases
with strong winds at Lake Abaya, is given in Weiß (2021). Out of these 20 strong wind events, two representative cases were
selected and investigated in this study.

### 2.2   ERA5 reanalysis

In addition to the measurements that allowed for the local perspective, the fifth generation of reanalysis data provided by the
European Center for Medium-Range Weather Forecasts (ECMWF), the ERA5 data set, was used (Hersbach et al., 2019). The



ERA5 data depicts the synoptic and mesoscale conditions as well as the vertical structure of the atmosphere prior and during strong wind evens at Lake Abaya. The hourly ERA5 data has a horizontal mesh size of $0.25° \times 0.25°$, which corresponds to roughly 28 km. Data on 37 pressure levels between 1000 and 1 hPa were used, which seemed sufficient for this work. Moreover, the ERA-Land subset was used which is a surface data set, forced by the ERA5 atmospheric parameters. The hourly ERA-Land data has a horizontal mesh size of $0.1° \times 0.1°$, which corresponds to roughly 9 km.

Since topography is smoothed in global models like the ERA5 reanalysis data, valleys and mountains are only poorly resolved, if at all. The ERA5 topography captures large-scale terrain features like the Ethiopian Highlands as a whole, the Afar Triangle at the northeastern or the Turkana Channel at the southwestern end of the GRV. The GRV with Lake Abaya is represented as well. However, the width of the valley near Lake Abaya is only about 50 to 100 km from crest line to crest line (see Fig. 1c) and, hence, only covered by a few ERA5 grid points. Therefore, the area around Lake Abaya is only poorly resolved in the ERA5 model topography. ERA5 topography near Lake Abaya is between 100 and 500 m higher than reality. The ERA5 grid point representing Lake Abaya best is at 6.5° N and 38.0° E with an elevation of 1317 m a.m.s.l., which is 142 m higher than the real elevation of the lake surface of 1175 m a.m.s.l. This grid point (LA), amongst others selected and used in the case studies, is marked in Fig. 1b–c.

## 2.3 WRF model

The Weather Research and Forecasting (WRF) model, specifically the Advanced Research WRF (ARW) version 4.1 (Skamarock et al., 2019), was used to conduct a numerical simulation for each of the two cases. The aim of using WRF was to resolve the flow in the valley better in space and time than the ERA5. Therefore, the simulation strategy involved two one-way, online nested domains (Fig.1a). The coarse domain (d01) included large parts of East Africa and had a horizontal grid spacing of $\Delta x = 3$ km with $750 \times 750$ grid points. To minimize the impact of spurious noise at the lateral boundaries of d01 on the nested domian d02, d01 had a three times larger geographical extent and served mainly as an intermediate step to produce boundary conditions for the inner domain (d02). This inner domain covered Ethiopia and particularly the main target area of the GRV with Lake Abaya as well as parts of the Afar Triangle and of the Turkana Channel, respectively (Fig. 1b). It had a horizontal mesh size of $\Delta x = 1$ km and again $750 \times 750$ grid points. The integration time step was 12 s in d01 and 4 s in d02.

A hybrid sigma-pressure coordinate (Park et al., 2019) was used with 80 vertical mass levels in both domains. The top of the model domain was located at 20 hPa which corresponds to about 26 km. The lowest level was approximately 20 m a.g.l. Close to the ground the vertical level spacing was $\Delta z = 20$ m which was stretched with increasing height to a maximum of 400 m. A damping layer after Klemp et al. (2008) was included to prevent wave reflection of vertically propagating gravity waves. This layer acted in the uppermost 10 km. For numerically stable simulations, it was crucial to cover the whole troposphere which reaches up to about 16 km in tropical regions. Hence, the damping layer acted only in the stratosphere.

WRF offers a variety of physical parametrizations. The setup for both domains is summarized in Table 1. In terms of convection, Yu and Lee (2010) found an upper bound for a convection-permitting grid spacing of 3 km, which is equal to the mesh size of domain d01. Hence, no cumulus parametrization was used in both model domains. In terms of land cover, the moderate-resolution imaging spectroradiometer (MODIS) 30 s ($\approx 1$ km) land use data set with a inland water body class was



**Table 1.** Summary of physical parameterizations used in both domains.

| Parametrization | Scheme | Reference |
|---|---|---|
| Microphysics | WRF Single-moment 6-class (WSM6) | Hong and Lim (2006) |
| Radiation | Rapid Radiative Transfer Model (RRTMG) | Iacono et al. (2008) |
| PBL | Mellor-Yamada Nakanishi Niino (MYNN Level 2.5) | Nakanishi and Niino (2009) |
| Surface Layer | Eta Similarity Scheme | Janjic (2002) |
| Land Surface | Unified Noah Land Surface Model | Tewari et al. (2004) |

used. The WRF default topographical data set, provided by the United States Geological Survey (USGS), with a resolution of 1 km was taken for both domains.

As initial and boundary conditions for d01 the operational high-resolution analysis (HRES) of the Integrated Forecasting System (IFS) of the ECMWF was applied. The HRES data set has a grid spacing of $0.1° \times 0.1°$ and is available for every six hours.

In terms of lake surface temperatures, the alternative initialization procedure provided by WRF was used where the daily averaged surface temperature was assigned as lake temperature to the grid points classified as inland water bodies. A comparison of the resulting lake temperatures to the Copernicus lake temperature data set, observed with satellites, revealed deviations of 1–2 K for different lakes in the domain. This difference was assumed to have no significant influence on the simulation results.

## 3   Synoptic conditions

This section illustrates the large-scale conditions prior to the two selected cases of strong winds in the GRV. The first case took place in the night from 14 to 15 January 2020 and was associated with winds from the northeast at Lake Abaya. The ITCZ was located south of Ethiopia. In the days before the event, the associated north and northeasterly trade winds advected cold air towards the Afar Triangle which is located at the northeastern end of the GRV (cf. Fig. 1a, 'AT'). This cold air advection was restricted below 800 hPa and blocked by the Ethiopian Highlands (Fig. 2a). Therefore, the depth of the colder air increased

in the days before the event. In the Turkana Channel and the region around Lake Abaya, hence, the at the southwestern end of the GRV, potentially warmer air was advected. This differential air mass advection led to an along-valley pressure which is depicted in Fig. 2a by the isolines of geopotential height $Z$ at the 800 hPa surface. 800 hPa roughly corresponds to the height of the pass near the city of Hawassa (cf. PA in Fig. 1b). The pressure gradient helped to channel the potentially colder air (higher pressure) from the Afar Triangle through the GRV towards the warmer air (lower pressure) around Lake Abaya and the

Turkana Channel. Hence, the strong northeasterly winds at Lake Abaya were associated with a gap flow through the GRV.

The second case took place on 5 June 2018 and was accompanied by strong winds from the south and southwest at Lake Abaya. The ITCZ was aligned along the northern edge of the Ethiopian Highlands. The Afar Triangle was therefore on the northern side of the ITCZ while the Turkana Channel was located south. Again, this led to differential air mass advection at lower tropospheric levels. Potentially warmer air from northeastern Africa was advected towards the Afar Triangle, whereas



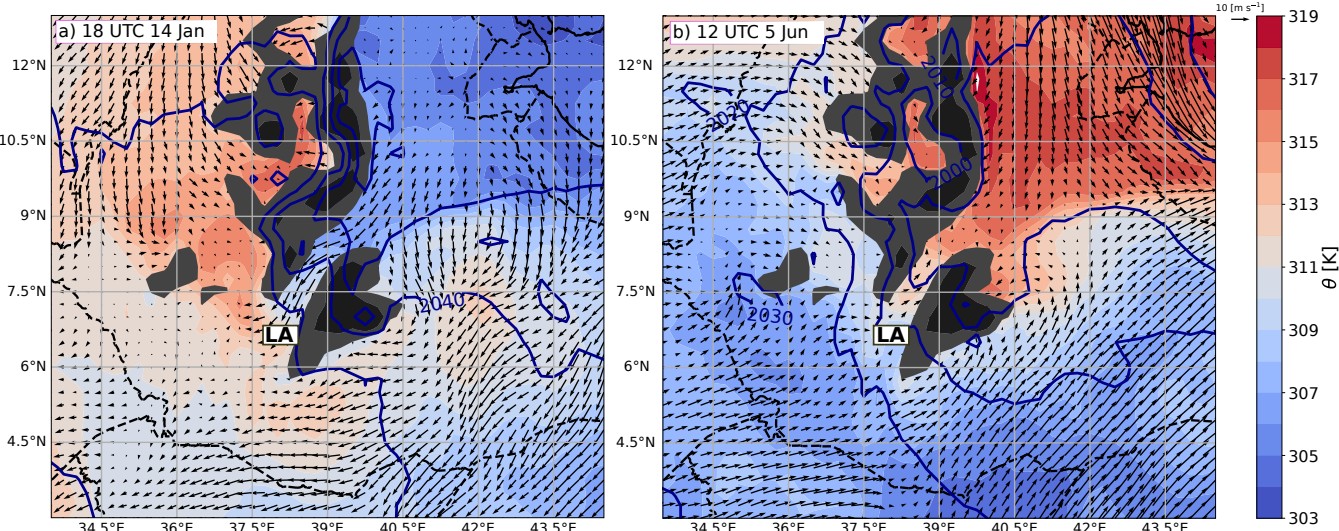

**Figure 2.** Potential temperature (color contours, 1 K increments), wind (arrows) and geopotential height (blue lines, 10 m increments) at 800 hPa in ERA5 over Ethiopia and Lake Abaya (LA) for (a) 18 UTC 14 January 2020 and (b) 12 UTC 5 June 2018. ERA5 model topography higher than 2000 m a.m.s.l. is shaded in dark grey.

the Turkana Jet (Nicholson, 2016) brought potentially colder air from the Indian Ocean into the Turkana Channel and the region around Lake Abaya (Fig. 2b). The different air masses north and south of the GRV established an along-valley pressure gradient, illustrated in Fig. 2b. Hence, the synoptic forcing in terms of the along-valley pressure gradient was in the opposite direction than in the first event but also caused a gap flow along the GRV. These southwesterly gap winds allowed potentially colder air in the Turkana Channel to advance towards the lower pressure north of the valley. Besides this pressure-driven

channeling (Kossmann and Sturman, 2003), however, the large-scale flow is non-negligible in the Turkana Channel and the Afar Triangle in this case (Fig. 2b). In fact, this large-scale flow caused inflow and, hence, forced channeling at both valley entrances.

## 4 Northeast gap flow

This section presents the northeast gap flow event. First, more insights about the development of the lower atmosphere prior to

the gap flow are given, followed by a description of the temporal evolution of the gap flow itself. The dependence on the time of the day of the strongest winds in the target area at Lake Abaya plays a special role.

### 4.1 Evolution of the main forcing

The pressure difference between both valley ends describes the magnitude of the larger scale forcing of the gap winds in the GRV and their timing. For this purpose, two ERA5 grid points were taken into account. The upstream grid point is in the Afar

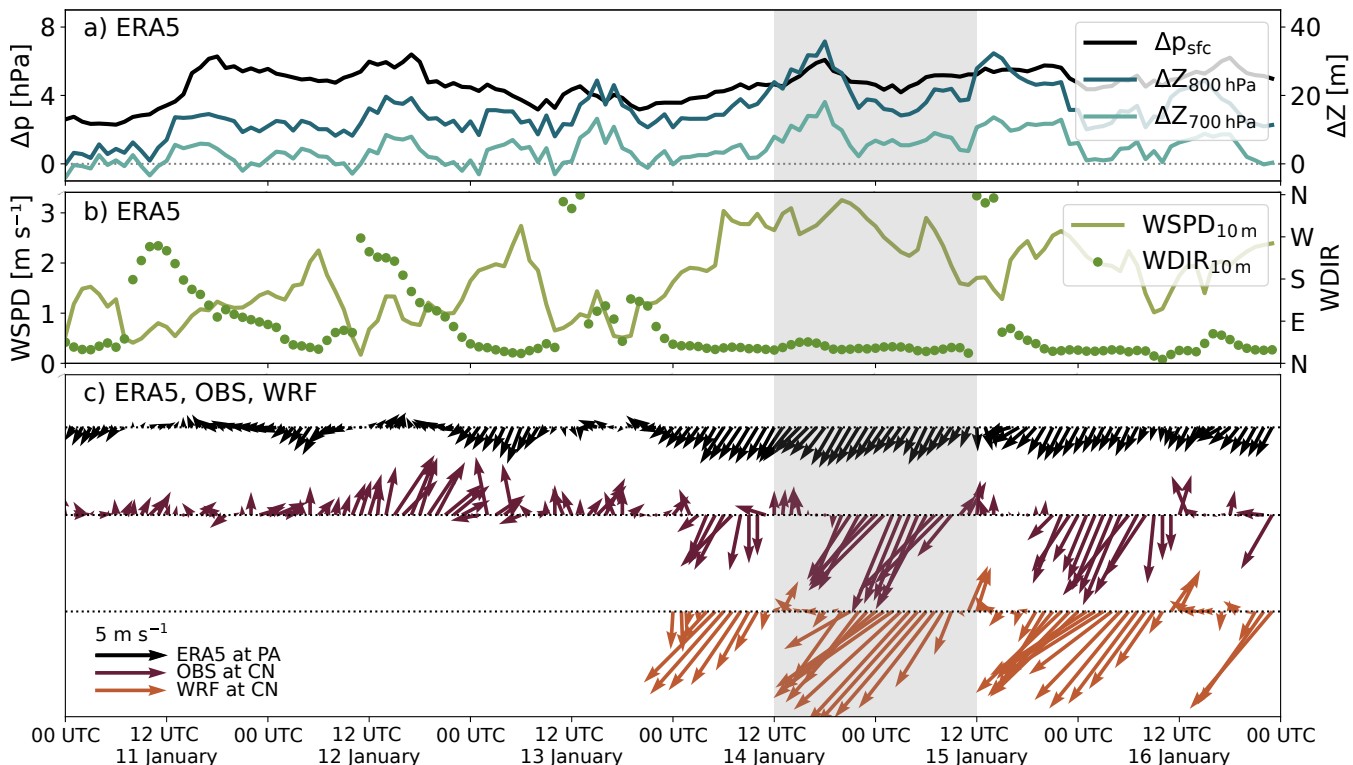

**Figure 3.** Time series from 11 to 16 January 2020 of (a) difference between two ERA5 grid points (Fig. 1b), one in the Afar Triangle (AT) and one in the Turkana Channel (TC), of surface pressure $\Delta p_{\mathrm{sfc}}$ reduced to 725 m a.m.s.l. as well as of geopotential height $\Delta Z$ at 800 and 700 hPa. (b) ERA5 10 m wind speed (WSPD) and wind direction (WDIR) for a grid point close to the pass (PA, also Fig. 1b,c). (c) Wind arrows for the same ERA5 grid point as in (b), as well as observations and simulated WRF winds for Chamo North (CN, also Fig. 1b,c) further downstream. The period of the investigated gap flow event on 14–15 January 2020 is shaded in grey.

Triangle (AT) at 10.25°N and 41°E and the downstream grid point is located in the Turkana Channel (TC) at 5°N and 37°E (cf. Fig. 1b). The pressure difference at a specific height was calculated as $\Delta p = p\,(\mathrm{AT})$ - $p\,(\mathrm{TC})$. Fig. 3 shows the times series of the difference in pressure reduced to 725 m a.m.s.l., $\Delta p_{\mathrm{sfc}}$, as well as the near-surface winds within the GRV. The considered gap flow event is highlighted by grey shading. The winds in the ERA5 near the pass (Fig. 3b) slowly but steadily intensified in the days before the actual gap flow event. A daily cycle in wind speed and wind direction is visible between 11–13 January with

southerly upvalley winds during the day and northeasterly downvalley winds during the night. On 14 January, the winds were strongest (but generally still rather weak) and remained northeast indicating flow through the valley from the same direction throughout the whole day. Wind vectors for the same ERA5 grid point are also shown in Fig. 3c. In addition, the flow at Chamo North is depicted, which is located further downstream at the northern edge of Lake Chamo (cf. Fig.1c). The wind arrows are based on AWS observations and the WRF simulation. Winds at Chamo North were generally stronger and revealed a different

timing than at the pass. Moreover, the strongest winds were measured and simulated in the night after 00 UTC (i.e. after 03


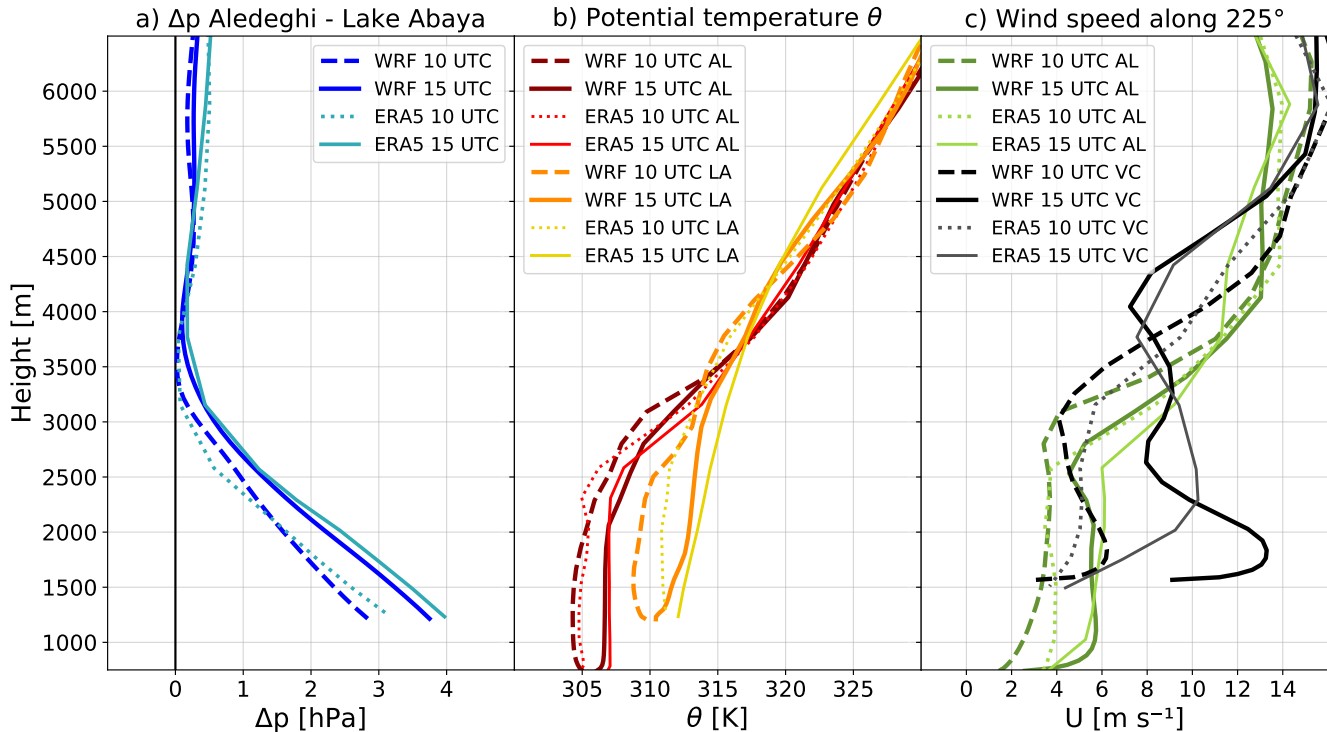

**Figure 4.** Vertical profiles at two model grid points at Aledeghi (AL) and Lake Abaya (LA, Fig. 1b), respectively, for both the WRF simulation and the ERA5 reanalysis. (a) Pressure differences $\Delta p$ in WRF (dark blue) and in ERA5 (light blue) for 10 UTC and 15 UTC 14 January 2020. (b) Potential temperature $\theta$, indicating the vertical stratification of the atmosphere up- and downstream, for both models at the two times. (c) Wind component in an azimuth direction of 225° (i.e. along the valley) for Aledeghi and for a model grid point in the along-valley center (VC, Fig. 1b).

East African Time, EAT) whereas winds peaked at the pass before 00 UTC. The observed and simulated wind pattern in Fig. 3c is similar in the night before and after the considered event which indicates that the gap flow was a recurring phenomenon spanning several consecutive days.

The difference in surface pressure $\Delta p_{\text{sfc}}$ in Fig. 3a cannot fully explain the timing of the gap flow event. It was at a comparable level already three days before the strongest winds at Lake Abaya. It mirrors the differential air mass advection below pass height and, hence, is mostly affected by the horizontal difference in the layer-mean temperature below 800 hPa up- and downstream of the pass. This is supported by the fact that the difference between the up- and downwind grid point (AT and TC) in geopotential height $\Delta Z$ around pass height (800 hPa) and crest height (700 hPa) are only marginally affected in the beginning (cf. Fig. 3a). However, later the increase of the depth of the colder air upstream in the Afar Triangle also increased the pressure gradient at pass height that peaked in $\Delta Z_{800\,\text{hPa}} > 30\,\text{m}$. This peak occurred on the day with the strongest winds in GRV and the target area.





Figure 4 provides vertical profiles of ERA5 and WRF at two grid points in the valley up- and downstream of the pass at 10 and 15 UTC 14 January. Notice, that the horizontal distance between these tow grid points, Aledeghi (AL) and Lake Abaya (LA), is shorter than the distance between the grid points used to derive the pressure and geopotential height differences in Fig. 3a (cf. Fig. 1b). The pressure difference over the shorter distance (Fig. 4a) below 3000 m a.m.s.l. increased between 10 and 15 UTC in both, ERA5 and WRF. At pass height ($\approx$ 2000 m a.m.s.l.) it increased in the WRF simulation from about 1.5 hPa to 2.2 hPa. The magnitude and vertical structure is very similar in WRF and ERA5.

The potential temperature profiles (Fig. 4b) show generally warmer air at Lake Abaya than at Aledeghi for both models. The temperature difference increased over the day for each up- and downstream pair and is consistent with the increase in the pressure difference $\Delta p$. For both models and both locations the boundary layer warmed during the day until sunset at 1533 UTC (1833 EAT) and exhibited a mixed convective boundary layer (CBL). While at 10 UTC the CBL was slightly lower on the downstream side in both, ERA5 and WRF, that changed at 15 UTC. At this time, the CBL reached up to more than 3000 m a.m.s.l. around Lake Abaya. Hence, the different evolution of the boundary layer and stronger warming on the downstream side in the target area compared to the upstream side enhanced the existing along-valley pressure gradient.

Fig. 4c shows the wind profiles for Aledeghi and a grid point in the along-valley center (VC, Fig. 1b). Depicted is the northeasterly wind component, i.e., the wind speed along the valley axis of 225°. The Aledeghi profile is characterized by a homogeneous wind speed in the CBL except near the surface due to friction. Above the CBL inversion the winds strongly increase. Between 10 and 15 UTC, the inflow and the northeasterlies in the valley increased for both locations. The increase at VC, however, was stronger with wind speeds of up to 10 m s$^{-1}$ in ERA5 and 13 m s$^{-1}$ in WRF. Furthermore, it exhibited a jet-like structure which was more pronounced in WRF than in ERA5

## 4.2 Evolution of the flow in the valley

Figure 5 illustrates the evolution of the northeast gap flow in the GRV based on simulated wind speed and potential temperature along the vertical transect a-b-c (see Fig. 1b) as well as the wind field at 2000 m a.m.s.l. The gap flow began shortly before sunset at around 15 UTC (Fig. 5a–b). A prominent gap flow feature is the continuous acceleration and the associated descending isentrops as the flow passed the narrowest section near Lake Ziway (see LZ in Fig. 5 a–b) of the GRV. The CBL was about 500–1000 m deeper downstream of the pass (Turkana Channel; point b) than upstream (Afar Triangle; point a), but exceeded pass height at both locations by at least 500 m (Fig. 5a). On the downstream side, the slight counteracting southwesterly (upvalley) flow at the northern shore of Lake Abaya was presumably the result of a combined lake breeze and slope/valley wind. Notice that this upvalley flow cannot be seen at 2000 m a.m.s.l. (Fig. 5b). After sunset the gap flow intensified, crossed the pass and reached the northeastern shore of Lake Abaya at 18 UTC (Fig. 5c–d). The horizontal wind component along the transect exceeded 24 m s$^{-1}$. Lake Abaya was not yet affected by the gap flow but rather by an opposing outflow from a side valley that terminates near the middle of the lake from southeast into the GRV (see Fig. 1c). The convergence of these two flows resulted in a hydraulic jump-like feature at LA (Fig. 5c).

Figure 6 shows the evolution of the gap flow near Lake Abaya and Lake Chamo along the shorter transect b'-c' (see Fig. 1b). In the early afternoon at 12 UTC (15 EAT) a lake breeze at the shores of the two lakes, especially at the northern shore

**Figure 5.** WRF simulation (domain d02) of the northeast gap flow on 14 January 2020 at (a)–(b) 15 UTC and (c)–(d) 18 UTC: (a), (c) Vertical transect along line a-b-c (see Fig 1b) through the GRV from the Afra Triangle (left) to the Turkana Channel (right) showing the potential temperature (black contour lines, 1 K increments), the horizontal wind component along the transect (color contours, 2 m s$^{-1}$ increments), and arrows for the 2D wind field on the vertical plane. (b), (d) Wind field at 2000 m a.m.s.l. with the horizontal wind speed (color contours, 2 m s$^{-1}$ increments) and wind barbs (half barb for 5 m s$^{-1}$ and full barb for 10 m s$^{-1}$). Grey shading represents the topography. In (c), (d) only the terrain exceeding 2000 m a.m.s.l. is shown. The bend of the transect at point b is marked in (a) and (c) by a vertical dashed line. Black lines in (b) and (d) indicate lakes resolved by the model. The following locations are marked: Aledeghi (AL) Lake Ziway (LZ), the along-valley center (VC) the pass (PA), the northern end of Lake Abaya (LA), AWS Alge (AG), God's Bridge (GB) and the AWS Chamo North (CN).





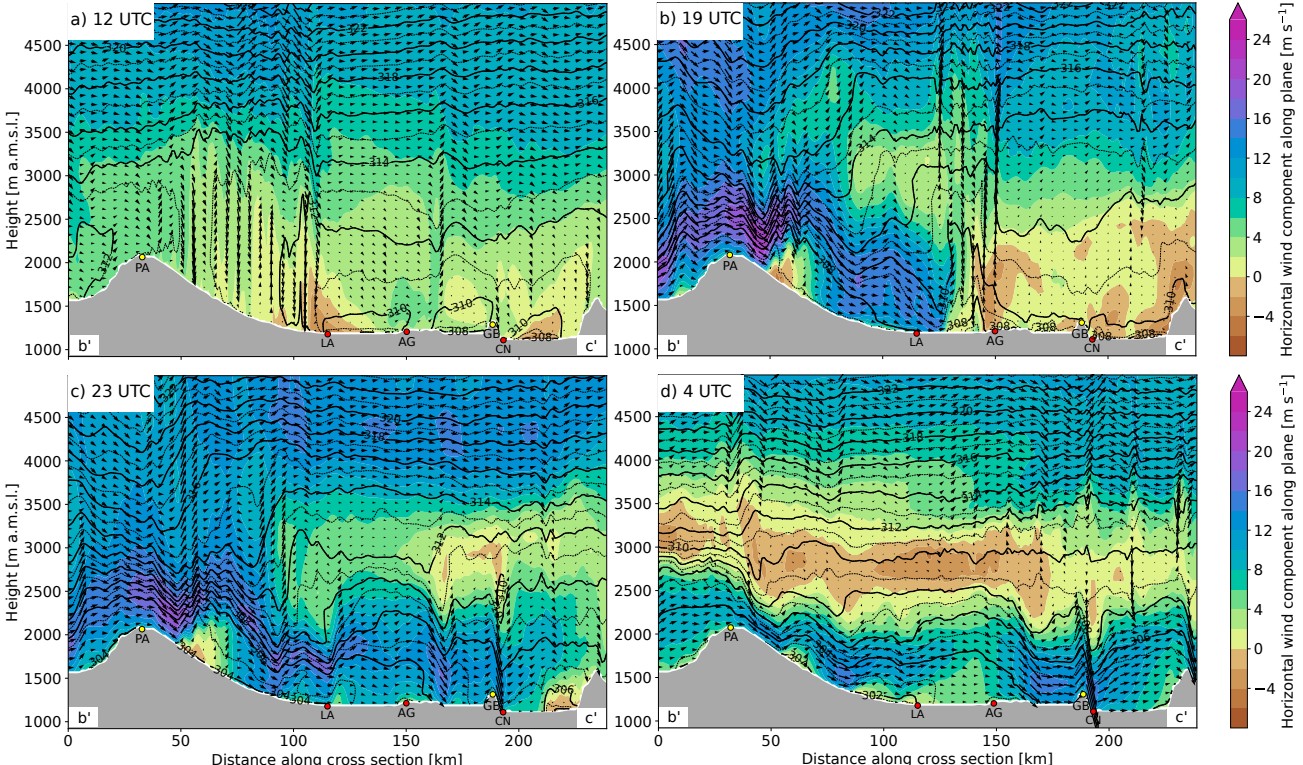

**Figure 6.** WRF simulation (domain d02) of the northeast gap flow: Vertical transect along line b′-c′ (see Fig. 1b) south of the pass (at 30 km) around Lake Abaya and Lake Chamo at: (a) 12 UTC, (b) 19 UTC, (c) 23 UTC 14 January 2020 and (d) 4 UTC 15 January 2020. Shown is the potential temperature (black contour lines, 1 K increments), the horizontal wind component along the transect (color contours, 2 m s⁻¹ increments), and arrows for the 2D wind field on the vertical plane. The pass (PA) and the northern edge of Lake Abaya (LA), AWS Alge (AG), God's Bridge (GB) and AWS Chamo North (CN) are marked with black dots. Grey shading represents the topography.

of Lake Abaya indicated by southwesterly winds, can be identified (brown colors near LA in Fig. 6a). On the slope between Lake Abaya and the pass convection took place and formed a deep convective boundary layer. Southwesterly upslope winds were weak but intensified three hours later (Fig. 5a). However, at the same time the gap flow established and prevented a further extent of the slope winds towards the pass. Due to the intensification of the gap flow after sunset the hydraulic jump

225 (i.e. the leading edge of the gap flow) propagated between 19 UTC (22 EAT) and 23 UTC (2 EAT) across both lakes and led to strong northeasterly winds over the lakes exceeding 10 m s⁻¹ (Fig. 6b–c). These winds were presumably strong enough to generate surface waves on Lake Abaya through wind stress. About 20 km downstream of the pass, the gap flow detached from the surface and formed a low-level rotor indicated by the reversed flow at low levels. The rotor remained for several hours (Fig. 6b-d). Vertical isentropes and weak winds on top of the gap flow near LA are indicative of gravity wave breaking. Local

230 topographic features induced local accelerations of the gap flow. This is depicted in Fig. 6c by the descending isentropes over





the southern part of Lake Abaya due to a local narrowing of the valley (cf. Fig. 1c) and over the northern part of Lake Chamo
due to Yegzier Dildiyi, the shallow hill between Lake Abaya and Lake Chamo. This local acceleration presumably explains the
strong winds at the AWS Chamo North. Until sunrise, near-surface winds remained strong but the gap flow became shallower
(Fig. 6d).

## 5  Southwest gap flow

In this section, the southwest gap flow event in the GRV on 5 June 2018 is described. The structure is the same as in Sect. 4:
starting with the development in the lower atmosphere followed by the description of the temporal evolution of the gap flow.

### 5.1  Evolution of the main forcing

Similar to the northeast gap flow event, the large-scale forcing is illustrated in Fig. 7a by the difference in pressure and geopo-
tential height between an ERA5 grid point upstream and downstream of the GRV. The Turkana Channel (TC) is now located on
the upstream side and the Afar Triangle (AT) on the downstream side, hence, $\Delta p = p\,(\text{TC}) - p\,(\text{AT})$ and $\Delta Z = Z\,(\text{TC}) - Z\,(\text{AT})$,
which is vice versa to the first case (cf. Fig. 3a). The positive surface pressure difference $\Delta p_{\text{sfc}}$ indicates that the pressure
was again higher upstream than downstream of the GRV. It continuously increased in the days before the investigated event.
The pressure gradient at pass height, depicted by the geopotential height difference $\Delta Z_{800\,\text{hPa}}$ increased in line with $\Delta p_{\text{sfc}}$ and
reached a maximum of about 30 m during the strongest winds near Lake Abaya, which is similar to the northeast gap flow. The
large-scale forcing was strongest within the GRV below crest height since $\Delta Z_{700\,\text{hPa}}$ remained small, oscillating around 0 m.

The ERA5 grid point near the pass exhibits southerly or southwesterly near-surface winds for most of the 7-day period
shown in Fig. 7b. They were strongest in the afternoon and weakest during the night, indicating a diurnal cycle and, hence,
a dependence on solar radiation. In addition, the persistent large-scale pressure gradient favored a long-lasting gap flow over
several consecutive days. Wind observations at Lake Abaya, in this case taken from the AWS at Alge (cf. Fig. 1c) show a
similar diurnal cycle as the ERA5 winds at the pass. The winds simulated by WRF at Alge also exhibit a diurnal cycle, but
were more than twice as strong than those observed. This is presumably due to lower friction associated with the underlying
*water* land use class at the WRF grid points surrounding Alge.

Figure 8a shows vertical profiles of the pressure gradient $\Delta p$ across the pass between Lake Abaya (upstream) and Aledeghi
(downstream). The pressure gradient $\Delta p$ was positive below main crest height (approx. 3000 m a.m.s.l.) and increased from
crest height to the height of the lake. The values are in the same range as for the northeast gap flow case, with about 2 hPa at
pass height. The horizontal pressure gradient slightly increased from 10 UTC to 15 UTC by about 0.5 hPa (Fig. 8a).

Figure 8b shows the evolution of the potential temperature up- and downstream of the pass. At pass height (about 2000 m
a.m.s.l.) potential temperature was about 8–10 K colder above Lake Abaya (upstream) than above Aledeghi (downstream of
the pass). Except from a shallow layer near the surface, the valley atmosphere at Lake Abaya was stably stratified, while a
deep CBL formed at Aledeghi profiles up to about 4000 m a.m.s.l. The temperature contrast between the up- and downstream
side slightly increased between 10 UTC and 15 UTC in both ERA5 and WRF, which is in agreement with the slight increase



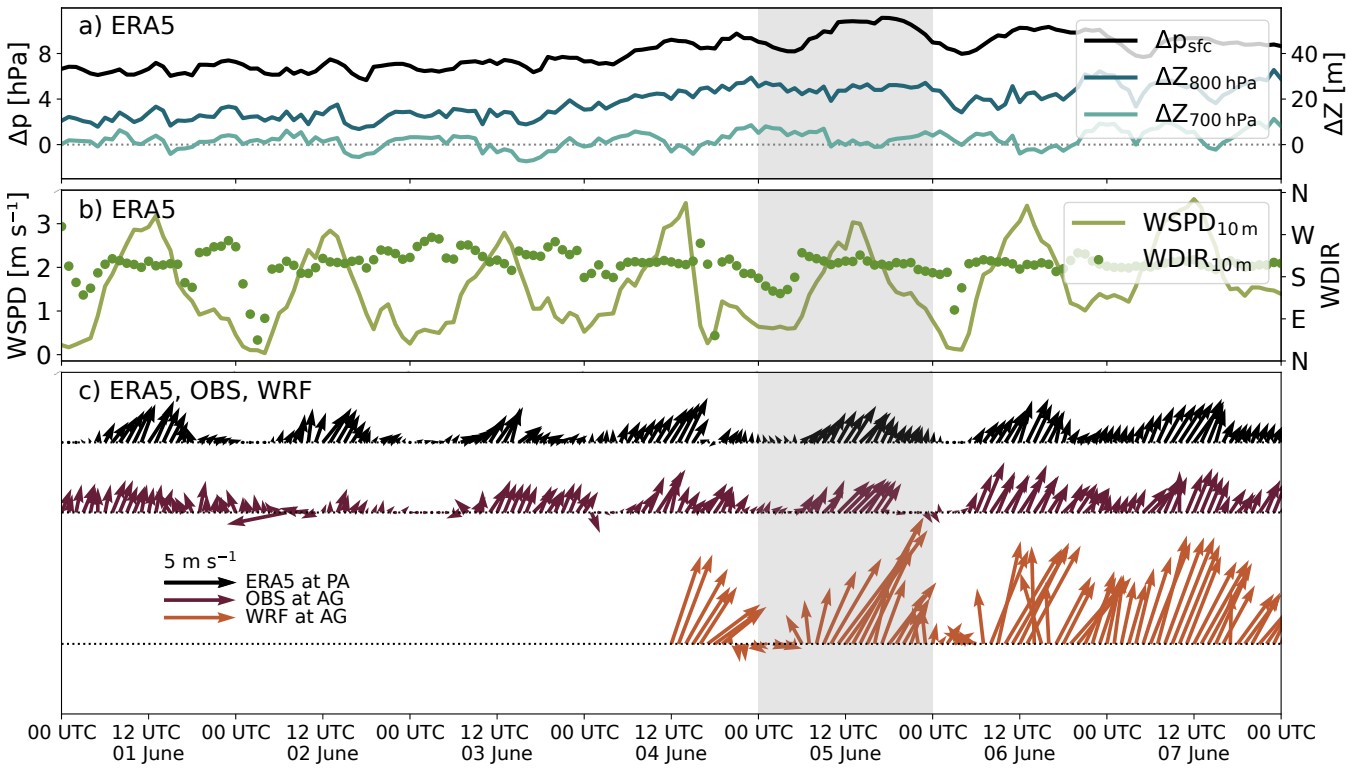

**Figure 7.** As in Fig. 3 but from 1 to 7 June 2018. Observations and simulated WRF winds are for the location Alge (AG, Fig. 1b,c). The period of the investigated gap flow event was on 5 June 2018 is shaded in grey.

in the pressure gradient. Below crest height, the WRF profile exhibited a multi-layer structure in temperature which was not captured by ERA5. Large differences can also seen in the wind profile at Lake Abaya (Fig. 8c), especially at 15 UTC with a

much stronger low-level jet in the WRF (up to 15 m s$^{-1}$) than in the ERA5 (up to 5 m s$^{-1}$). At the grid point of the along-valley center (VC), the discrepancy smaller but with a difference in near-surface wind speed of 5 m s$^{-1}$. Hence, ERA5 captured the basic feature of the gap flow, including the increase in wind speed between 10 UTC and 15 UTC, but again underestimated the magnitude of the gap winds, similar to the first case (cf. Fig. 4c and Fig. 8c). An interesting feature is the wind reversal near crest height, as indicated by the change of sign of the along-valley component in Fig. 8c. Hence, in contrast to the first case

(Fig. 4c), the gap flow in the valley seemed to be completely decoupled from the synoptic flow above crest height.

## 5.2 Evolution of the flow in the valley

The development of the southwest gap flow in the GRV is illustrated by the vertical transect a-b-c in Fig. 9a,c and the wind field at 2000 m a.m.s.l. in Fig. 9b,d. Notice that the color scale for the along-valley wind component in Fig. 9a,c is reversed compared to Fig. 5a,c in order to highlight the reversed gap flow with the same bluish colors. The same is the case for the

shorter cross sections a′-b and b′-c′ shown in Fig. 10.



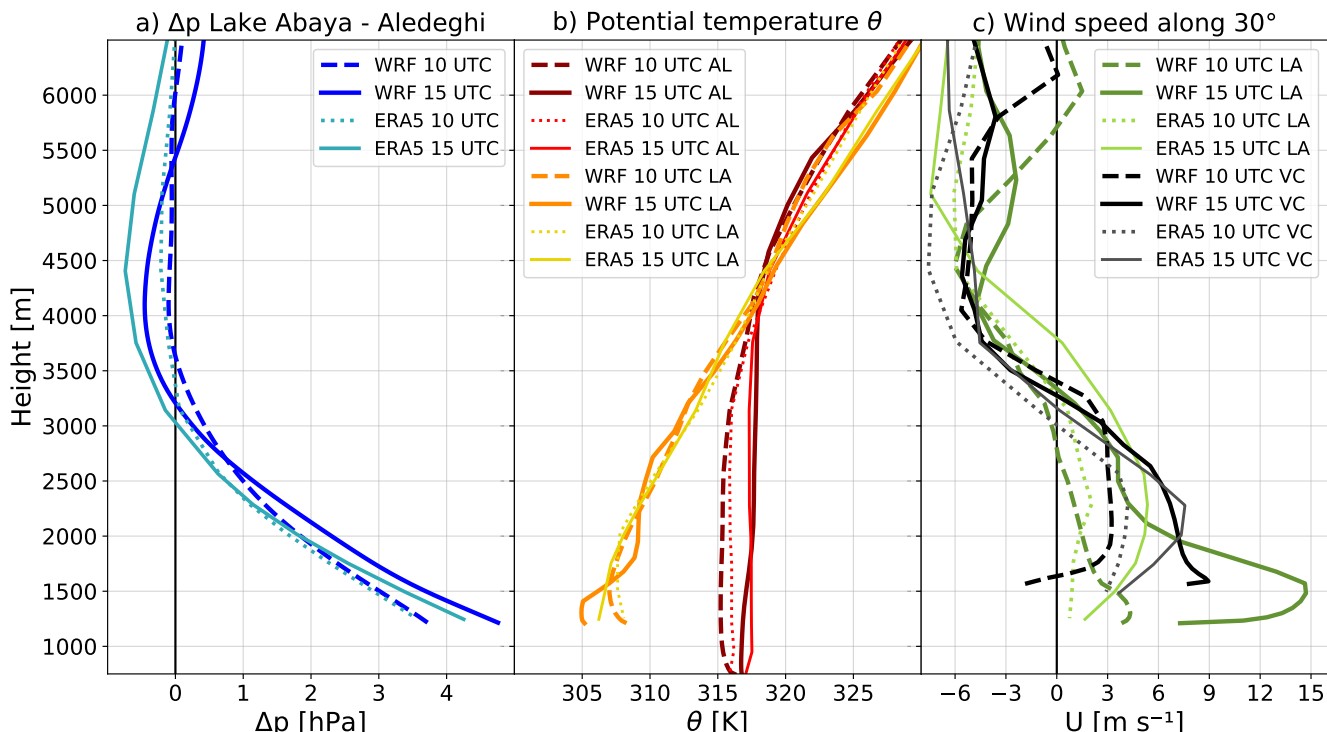

**Figure 8.** As in Fig. 4 but for 10 UTC and 15 UTC 5 June 2018. The pressure difference $\Delta p$ is calculated as the pressure at Lake Abaya (LA) minus the pressure at Aledeghi (AL; Fig. 1b). The wind profiles for Lake Abaya and the along-valley center (VC) in (c) refer to the wind component in an azimuth direction of 30°, i.e. along the valley in northeasterly direction.

In the late afternoon of 5 June at 14 UTC (17 EAT, Fig. 9a), the different structure of the boundary layer up- and downstream of the GRV was one of the most striking features. On the downstream side, in the Afar Triangle, the air mass was much warmer and exhibited a CBL deeper than 4 km a.m.s.l. (see also Fig. 8b). Resolved up- and downdrafts in the CBL, indicated by vertically pointing arrows, were associated with cumulus convection. The strong temperature contrast between the up- and downstream sides and the associated horizontal temperature gradient inside the valley is depicted by slanted isentropes (Fig. 9a). In the Turkana Channel and the region around Lake Abaya, the boundary layer was stably stratified due to cloud formation, except for a shallow mixed layer with a depth of about 200 to 300 m over Lake Abaya (see also Fig. 8b). The southwesterly flow into the valley, advecting colder air from the Turkana Channel, was restricted to the layer below 3500 m a.m.s.l. The structure of the bended isentropes upstream of the pass and the associated wind field showed similarities to a density current. The latter is characterized by an elevated head (cf. Fig. 2 in Gohm et al., 2010). This situation was persistent for roughly six hours, long enough to generate waves on the lake. Local flow acceleration illustrated in Fig. 10b between CN and LA was most likely caused by the hill between Lake Chamo and Lake Abaya (Yegzier Dildiyi) and by the narrowing valley near the center of Lake Abaya (see in Fig. 1c). This is similar to a hydraulic flow transition from a sub- to a supercritical state (e.g.,





**Figure 9.** As in Fig. 5 but for the southwest gap flow at (a)-(b) 14 UTC and (c)-(d) 19 UTC 5 June 2018. Notice the reversed color scale in (a) and (c) compared to Fig. 5a,c.

Pan and Smith, 1999). Strongest winds in the core of the low-level jet over Lake Abaya (at about 300 m a.g.l.) reached about
20 m s$^{-1}$ (Fig. 10b), but hardly exceeded 10 m s$^{-1}$ near pass height at 2000 m a.m.s.l. (Fig. 9b). At the northern shoreline, the
flow slowed down while ascending the slope towards the pass (Fig. 9a and Fig. 10b). North of the pass, the wind speed was
still weak, although mostly from southwesterly directions (Fig. 9a and 10a). This changed after sunset, similar to the northeast
gap flow case.

At 19 UTC, the flow accelerated downstream of the pass and also downstream of the narrowest valley section at Lake
Ziway (see PA and LZ in Fig. 9c,d). The acceleration was associated with descending isentropes and, hence, a reduction of
the flow depth. Both are again signs of a hydraulic transition into a supercritical state. The gap flow had reached the Afar

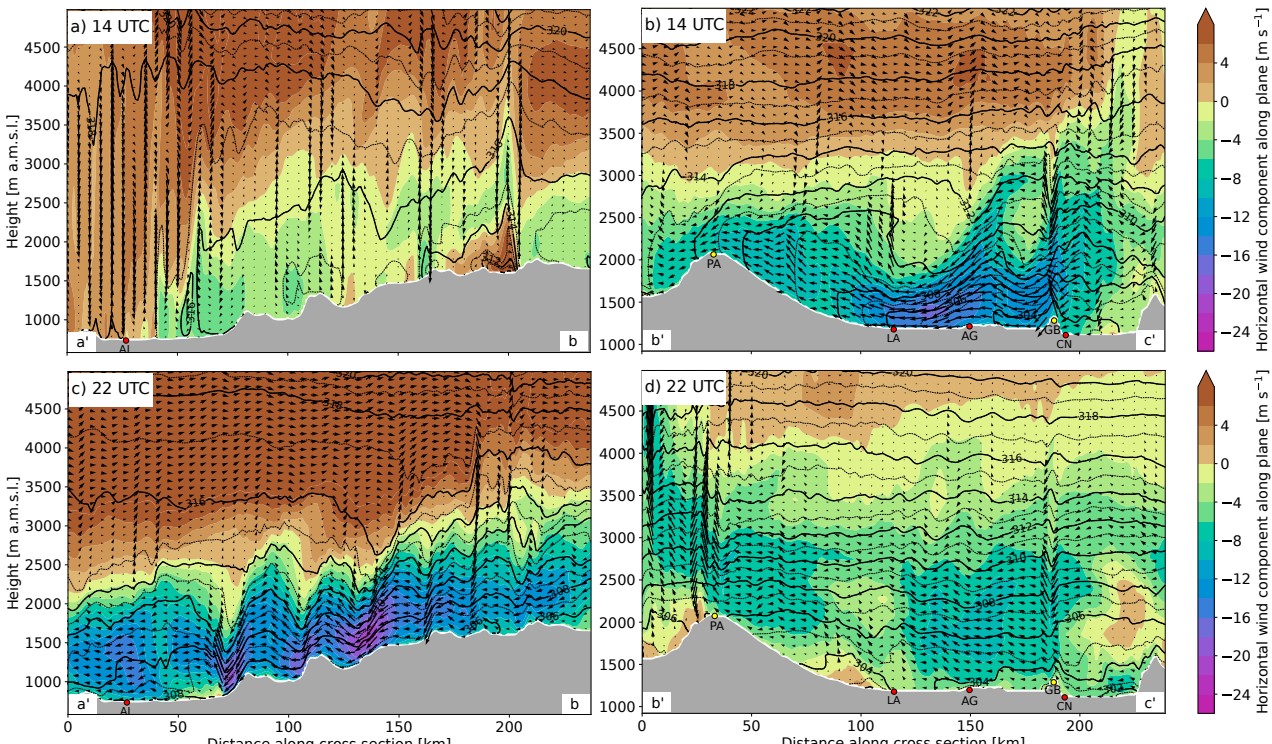

**Figure 10.** WRF simulation (domain d02) of the southwest gap flow: Vertical transect (a) and (c) along line a′-b (see Fig. 1b) in the northern part of the GRV and (b) and (d) along line b′-c′ around Lake Abaya and Lake Chamo at (a)-(b) 14 UTC and (c)-(d) 22 UTC 5 June 2018. Shown is the potential temperature (black contour lines, 1 K increments), the horizontal wind component along the transect (color contours, $2\,\mathrm{m\,s^{-1}}$ increments), and arrows for the 2D wind field on the vertical plane. The pass (PA), the northern edge of Lake Abaya (LA), AWS Alge (AG), God's Bridge (GB) and AWS Chamo North (CN) in the upstream section and Aledeghi (AL) in the downstream section are marked with black dots. Grey shading represents the topography. Notice the reversed color scale compared to Fig. 6

Triangle. Wind speeds were moderate (around $15\,\mathrm{m\,s^{-1}}$), but intensified later in the night reaching about $24\,\mathrm{m\,s^{-1}}$ north of the pass (see Fig. 10c). The evolution is similar to the northeast gap flow. Several hills embedded in the valley and lateral topographic constrictions led to local flow accelerations downstream of the pass (Fig. 10c). The passage of the gap flow "front"

removed the near-surface inversion that had formed in the Afar Triangle under the deep residual layer after sunset (Fig. 9c). The complete wind reversal at about $1500\,\mathrm{m}$ a.g.l. downstream of the pass (Fig. 10c) indicates that was gap flow was completely decoupled from the mid-tropospheric flow. After the initial phase of strong winds at Lake Abaya in the afternoon (Fig. 10b), wind speed decreased over the lake in the evening presumably due to partial flow blocking upstream of the pass (see, e.g., terrain-intersecting isentropes in Fig. 10d). Hence, in contrast to the northeast gap flow, wind speed was weak during the night

(Fig. 10d).





## 6 Discussion

The reported strong winds at Lake Abaya were the result of processes on different scales. Although the two investigated cases are characterized by flows from different directions, they have several characteristics in common. Both events can be classified as a gap flow phenomenon in the GRV, Ethiopia. Gap winds result from a hydraulic response of the flow to changes
in terrain height and valley width (e.g., Pan and Smith, 1999). As the flow in the GRV experienced lateral constriction due to a narrowing of the valley width and a rising valley floor, it accelerated and exhibited the typical asymmetric flow structure up- and downstream of the pass. Simulated wind speeds were up to $25 \, \mathrm{m \, s^{-1}}$, which are similar to other gap flows like *shallow foehn* (Gohm and Mayr, 2004; Haid et al., 2020), *shallow bora* (Gohm et al., 2008; Grisogono and Belušić, 2009) or *taku* (Bond et al., 2006). Topographic features like the hill Yegzier Dildiyi between Lake Abaya and Lake Chamo as well as the
lateral constriction near the center of Lake Abaya locally enhanced the gap flow in the target area. The same is true for other regions in the GRV characterized by changes in the valley width and floor height.

Preceding both cases, differential advection of air masses on the two adjacent lowlands established an along-valley pressure gradient and, hence, an important dynamical driver on the synoptic scale. The colder and more stably stratified air, and therefore the higher pressure, was always on the upstream side. For the northeast gap flow, the colder air mass advected into the Afar
Triangle upstream of the pass was blocked by the Ethiopian Highlands and piled up over a few days, similar to the cold-air blocking south of the Brenner Pass during shallow foehn in the European Alps (Zängl, 2002). The horizontal difference in geopotential height at $800 \, \mathrm{hPa}$ (i.e. near pass height) over a distance of about $800 \, \mathrm{km}$, $\Delta Z_{800 \, \mathrm{hPa}}$, exceeded in both cases a threshold of $30 \, \mathrm{m}$. Hence, this variable may be used as a predictor for forecasting gap winds in the GRV, similar to the large-scale predictors used by Drechsel and Mayr (2008) for forecasting *foehn* in the Wipp Valley, Austria. Further case studies and a
climatological analysis, however, are necessary to proof the applicability of this predictor. Besides the synoptic driver, however, both cases revealed a strong diurnal cycle, which highlights the importance of the thermal forcing due to differential heating over complex terrain. This thermal forcing acted at least on four different scales (e.g., De Wekker and Kossmann, 2015): (1) the two adjacent lowlands (Afar Triangle and Turkana Channel) causing very different CBL growths and enhancing the air mass contrast, (2) the Ethiopian Highlands representing a plateau that acts as an elevated heat source and/or the GRV representing
a basin, (3) the GRV being a valley with embedded passes and hills inducing valley and slope winds, and last but not least (4) Lake Abaya inducing lake and land breezes, that enhanced or weakened the gap wind or delayed its onset locally. The relative contributions of these forcings could not be disentangled in this study, but it is conceivable that the smaller scales did not determine but only modify the occurrence of gap winds. It is noteworthy, that the strongest gap winds in the GRV formed in the late afternoon and night when daytime convection decayed, indicating that convection may be a delaying mechanism
despite its importance for differential heating.

The biggest difference between the two cases of strong winds at Lake Abaya were their time of occurrence. The mature stage of the northeast gap flow occurred over Lake Abaya in the night and early morning long after convection had decayed. This is in line with eyewitness reports (T. Minda, personal communications), indicating that the northeasterly winds, the *Gidabo Winds*, occur mainly during nighttime and in the early morning. In contrast, the southwest gap flow was strongest at Lake Abaya





in the afternoon and weakened in the night due to partial flow blocking upstream of the pass. However, in the Afar Triangle downstream of the pass, the gap flow broke through in the evening and intensified over the course of the night. Therefore, the timing and intensity of the gap winds in the GRV strongly depend on the region of interest. It is, however, worth mentioning that in both cases the dynamic forcing was not able to form the gap winds alone. The thermal forcing was an important supporter, if not a key player, on the larger scale but presumably a delayer on the smaller scale. Hence, this study highlights the complex

interactions of different mechanisms on different scales.

Plateau and basin winds could have played dual roles, especially during the investigated northeast gap flow. On the one hand, the Ethiopian Highlands could have acted as an elevated heat source. The resulting inflow from the adjacent lowlands in form of a density current was suppressed until sunset when convection decayed in the Afar Triangle and the Turkana Channel (cf. Zängl and Gonzalez Chico, 2006). According to Doran and Zhong (2000), a major gap, like the GRV, in the mountain ranges

around a basin or in the plateau allows for (intense) inflow already in the afternoon which was the case for the northeast gap flow. The premature and mature states of this inflow are depicted in Fig. 5a and Fig. 5c, respectively. The inflow from the northeast was stronger than from the southwest, presumably because it was superimposed by the dynamically driven flow. On the other hand, the GRV itself could be interpreted as a narrow, elongated basin within a mountain range, similar to the Mexico City Basin (Doran and Zhong, 2000). The elevation of the valley floor varies only slightly over a distance of about 170 km (see

Fig. 5a between 300 and 470 km. The valley exhibited higher potential temperatures compared to the Afar Triangle northeast as indicated by descending isentropes (Fig. 5a) towards the elevated basin, i.e. the inner part of the GRV. The resulting circulation in Fig. 5a shows similarities to the basin and plateau circulation illustrated in Fig. 10 of Doran and Zhong (2000) and Fig. 10 of Zängl and Gonzalez Chico (2006). For the southwest gap flow, plateau winds on the scale of the Ethiopian Highlands may have influenced the gap flow in the valley. Such plateau winds are depicted in Fig. 9b at 2000 m a.m.s.l. by upslope

flows over the highlands resulting in flow convergence at the highest part of the terrain. Basin winds could only have played a counteracting part at the northeastern valley entrance that delayed the onset of the gap flow in the Afar Triangle. No strong evidence supporting this hypothesis, however, was found. The reversed flow in Fig. 9b at about 38.8° E and 8.7° N and in Fig. 10a at $x = 190$ km) is associated with outflow of a convective cell.

The effect of the Turkana Jet on the gap flow in the GRV is not fully clear. According to Nicholson (2016) this jet is strongest

in the late night and between June and September. On the one hand a partial deflection of the Turkana Jet into the valley may enhance the inflow into the GRV for southwest gap winds and on the other hand it may limit the southward extent of northeast gap winds. Additionally, a connection between the nocturnal low-level jet in northwestern Ethiopia and northeasterly gap flows in the GRV are imaginable (cf. similar flow pattern and temperature distribution in Fig. 2a and Fig. 12a in Rife et al., 2010).

Comparing the WRF model outputs with the ERA5 reanalysis data showed, that the investigated gap flows are represented

qualitatively well in the ERA5 model. The ERA5 profiles of pressure, temperature and wind in Fig. 4 and 8 showed generally a good agreement with the WRF profiles. There are, however, discrepancies in the height of the boundary layer top and especially in the strength of the low-level flow in the valley. The ERA5 winds were weaker (see Fig. 4c and 8c). This is most likely due to the coarser model topography with a smoothed terrain and less steep slopes. For instance, the elevation difference between the pass and Lake Abaya is 500 m in ERA5 and about 900 m in WRF d02. Local flow acceleration by embedded hills (e.g., Yegzier





Dildiyi), valley width changes and inflow from small side valleys are not resolved in ERA5. In contrast to wind speed in the valley, the inflow from the Afar Triangle into the valley for the northeast gap flow is nearly identical to the WRF simulation (cf. Fig. 4c).

Hydraulic theory explains gap winds by a transition from a sub- to a supercritical state (with respect to Froude number) at the narrowest and highest section of the terrain, resulting in a continuous flow descent and acceleration (e.g., Pan and Smith, 380   1999). This behavior is indicated for both events by the descending isentrops behind the strongest lateral constriction in the GRV around Lake Ziway, downstream of the pass and locally at Lake Abaya (e.g., Fig. 5c, Fig. 6c and Fig. 10b–c). Applying reduced-gravity shallow-water theory supported the expected transition from a sub- to a supercritical flow but is inaccurate due to the ambiguity in determining the channel width and the Froude number for a continuously stratified atmosphere (cf. Durran, 2015; Weiß, 2021).

**7   Conclusions**

The aim of this study was to reveal the cause and properties of strong winds at Lake Abaya and in the GRV in Ethiopia. Two cases, classified as a northeastern and a southwestern gap flow, where examined in detail based on measurements, ERA5 reanalysis data and mesoscale simulations conducted with the WRF model. The major findings are:

– In both cases, strong winds at Lake Abaya were caused by gap flows in the GRV. The valley exhibits a lateral constriction 390   northeast of Lake Ziway and a pass near the city of Hawassa that favor acceleration as the air flows through the valley from the Afar Triangle to the Turkana Channel or vice versa. Maximum wind speed in the WRF model was in both cases about $25\,\mathrm{m\,s^{-1}}$, however, not directly over the lake surface.

– Prior to both cases, different air masses were advected north and south of the GRV, respectively, with the colder and more stably stratified air mass on the corresponding upstream side. The differential air mass advection led to the formation of 395   an along-valley pressure gradient. The pressure gradient at $2000\,\mathrm{m}$ a.m.s.l. (about $800\,\mathrm{hPa}$), which corresponds roughly to the height of the pass, turned out to be more suitable for describing the timing of the gap flow than the formation of the surface pressure gradient over the lowlands.

– The strongest gap winds occurred downstream of the pass and the narrowest valley section. For the southwest gap flow Lake Abaya was located upstream of the pass. Nevertheless, in this case acceleration occurred also in the upstream 400   section and the local winds above Lake Abaya were enhanced by the hill Yegzier Dildiyi and the narrowing valley near the center of the lake. For the northeast gap flow Lake Abaya was located downstream of the pass and, hence, experienced stronger winds. Over the lee slope between the pass and the lake an atmospheric rotor as well as gravity wave breaking occurred in the simulation.

– Both cases exhibited a strong dependence on the time of the day which suggests that the gap winds where not only 405   dynamically driven but were strongly influenced by thermal forcing due to differential heating over complex terrain. However, the relative contribution of these forcings could not be quantified in this study. The onset of the gap flow near





the pass happened around sunset for both events. In the target area at Lake Abaya, the strongest wind occurred at night
for the northeast gap flow and in the afternoon for the southwest gap flow, respectively. This timing was mainly caused
by the up- or downstream position of Lake Abaya relative to the pass. Additionally, the northeastern flow opposed the
upvalley wind downstream of the pass during the day, whereas for the southwest case the gap flow pointed in the same
direction as the upvalley winds at Lake Abaya.

– Both events are represented in the ERA5 reanalysis data qualitatively well but gap winds are weaker in comparison to
the WRF simulation.

This work provides first insights into strong winds at Lake Abaya and the GRV based on two case studies. It illustrates

that these winds are the result of a complex interplay between different scales and forcings. Future research should focus on
compiling a climatology of such gap winds or on developing a simple forecasting tool similar to the approach of Drechsel
and Mayr (2008). Given the scarcity of data in this region, ERA5 could be a suitable dataset for such studies. Furthermore,
clarifying the role of the synoptic, thermal and topographic drivers could be addressed with numerical sensitivity studies.

*Code and data availability.* The WRF model (https://www.mmm.ucar.edu/weather-research-and-forecasting-model) and the ERA5 data set

(https://cds.climate.copernicus.eu) are publicly available. Observational data and codes for analysis are available on request from the authors.

*Author contributions.* CIW performed the simulations and the analysis and wrote the manuscript based on his Master's thesis. AG was his
main supervisor and contributed equally to the interpretation of results and the finalization of the manuscript. The idea for this study arose
on a boat trip on Lake Abaya of MWR, TTM and Fassil Teffera (colleague of TTM) during a visit of MWR in Arba Minch. MWR and TTM
were co-supervisors of the Master's thesis and contributed with proofreading.

*Competing interests.* The authors declare that they have no conflict of interest.

*Acknowledgements.* The AWS of the GEMS network were supplied in part by the Meteorology and Air Quality Group of Wageningen
University & Research and funded by Nuffic for the PhD project of TTM, and in part by the Arba Minch University Institutional University
Cooperation (AMU-IUC), funded by VLIR-UOS. AMU-IUC also provides financial support for the operation of the GEMS network. Further,
we like to thank Fassil Teffera for sharing his critical observations and knowledge with us. The computational results presented here have

been achieved (in part) using the LEO HPC infrastructure of the University of Innsbruck.



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
