# Peer review of "Dynamics of Gap Winds in the Great Rift Valley, Ethiopia: Emphasis on Strong Winds at Lake Abaya"

_Weather and Climate Dynamics, 2022_

## Referee Comment (RC1)

Review for

*Dynamics of Gap Winds in the Great Rift Valley, Ethiopia: Emphasis on StrongWinds at Lake Abaya*

by Weiss et al.

**Synopsis:**

The manuscripts present an interesting example of a gap wind in the Great Rift Valley, Ethopia, i.e., a region that attracted only little focus in atmospheric dynamics. Many dynamical mechanisms leading to the gap winds are discussed, e.g., the build-up of pressure difference along the value due to differential temperature advection, hydraulic flow behaviour due to valley slopes and constrictions, impact of convective activity,... I think the processes are well described, the figures of high quality, and the text is well structured and written in a clear, concise way. The only concern that could be brought up is that the evolution of the gap winds is discussed in a rather descriptive/qualitative way, without quantifying the forcing mechanisms. Still, it is a coherent story that is told about an interesting weather feature in a rarely studied region. Therefore, I can recommend publication of the study in *Weather and Climate Dynamics* if the following (mostly) minor comments are adequately addressed. In addition to physical aspects, I also make a few suggestions about wording.

**Comments:**

- L4: 'Therefore, two...' -> To this aim, two...'

- L15: 'were location dependent' -> 'depended on location'

- L17,18: 'The ERA5... Hence, ERA5 ... gap flows': I am not sure whether keep this more data-related aspect in this more mechanism-focused abstract.

- L20: 'produce water waves' -> 'induce surface waves on the lake'

- L29: 'The average depth of the lake is 8.6 m' -> It is interesting, although beyond the scope of this study, to think about how this rather shallow lake depth affects the waves that form on the surface. Possibly, the authors want to refer to one, two studies that address the modeling of surface waves on lakes, e.g.,

> *Seibt, Christian, Peeters, Frank, Graf, Michael, Sprenger, Michael, Hofmann, Hilmar, (2013), Modeling wind waves and wave exposure of nearshore zones in medium-sized lakes, Limnology and Oceanography, 58, doi: 10.4319/lo.2013.58.1.0023.*

*Graf, Michael, et al. "Evaluating the suitability of the SWAN/COSMO-2 modelsystem to simulate short-crested surface waves for a narrow lake with complex bathymetry." Meteorologische Zeitschrift 22.3 (2013): 257-272.*

The two studies illustrate how the ecology, but also archeological sites are affected by waves, in particular in the shallow shore region. I do not think that this wave-on-the-lake aspect is crucial for the study, but it is a nice point how strong winds can have indirect effects/impacts. If the introduction could be shortened, it would certainly be this relatively long paragraph (L27-50) about ecology and waves. However, as mentiond before, this is not mandatory from my side.

- L45: 'a bidirectional effect' -> 'two effects'

- L55,56: Avoid two consecutive sentences starting with similar 'connecting' words (Hence, Therefore)

- L52: 'Besides other large-scale drivers like ENSO...' -> How relevant is ENSO for *weather* in Ethopia? I am not sure that such a climate index is particularly relevant for this study. If yes, it could be discussed in somewhat greater detail

- L58: 'dual passage' -> 'biannual | twice-yearly passage'

- L69: In L49-50 it is written that the the waves coudl be the reason why Lake Abaya is less affected by invasive plants than, e.g., Lake Tana. In L45 it is argued that it is Lake Abaya's large size that allows many waves to evolve because of the large wind tech. On the other hand, Lake Tana is larger, offering a larger wind fetch. Wouln't one expect larger waves? Hence, is it more the wind speed that determines the wave acticvity than the wind fetch?

- L63: 'The adjustment of the wind field by local effects' -> What does this exactly mean? Could you be somewhat more specific!

- L66: 'shows strong differences to the synoptic flow' -> 'strongly deviates from the synoptic flow'

- L69: 'Lake Abaya is at least in some months connected to the large scale flow' -> Could you be a little more specific? What does 'connect' mean in this context?

- L73: 'on the lake water dynamics' -> 'on the lake's wave dynamics and internal mixing'

- L74: 'Therefore these winds...' -> 'It is the aim of this study to better understand these winds...'

- L76: 'the goal of' -> 'the more specific goal of'

- L79: 'The synoptic conditions prior to the two cases' -> Why *prior* to these events?

- L89: Remove 'that allowed for the local perspective'

- L92: 'horizontal resolution of 0.25 x 0.25' -> make clear that ERA5 is based on a spectral model and thus provide the spectral resolution, the grid spacing in latitude/longitude is 'only' an interpolation from spectral space

- L101: 'ERA5 model topography. ERA5 topography near...' -> 'ERA5 model topography, which near ...'

- L140: 'Lake Abaya, hence, the at the...' -> correct sentence structure!

- L141-143: 'led to an along pressure' -> more precisely, 'led to an along pressure gradient'

- L142: 'height Z at the 800 hPa surface. 800 hPa...' -> 'height Z at the 800 hPa surface, which...'

- L143: Here it is written that the pressure gradient *helped* to channel the air though the gap. The term 'helped' is somewhat too unspecific? Which other processes drive the air through the channel? To which degree does the pressure gradient contributes to the flow through the valley, and to which degree are other processes (which?) essential? I am quite convinced that the pressure gradient is indeed decisive, but the wording in the text could be more careful. In the same line of argument I wonder to which degree it is possible to understand the pressure gradient at 800 hPa by means of hydrostatic effects, i.e., due to the differences in the teperature?

- L155-157: Here, it is state that the large-scale flow is non-negligible, because it caused an inflow and hence forced channeling at both valley entrances. I am not sure whether I understand this point, and whether I see this inflow channeling in the figure. Possibly, my point is also related to the following question: Is the large-scale flow also associated with a pressure gradient that is perpendicular to the valley axis and thus contributes to the along-valley acceleration of the flow. In short, a more careful discussion/distinction of local effects and large-scale effects would be helpful.

- L162.163: The pressure difference...and their mixing' -> I had to read this sentence several times, and still I am not sure that I get it correctly. Please rephrase in clearer way. Possibly, a way to do so would be: 'Two points are selected at the valley entrance and exit, and the pressure difference between these two points is then take as a diagnostic for the gap winds in the GRV and their temporal evolution'.

- Figure 3 and the corresponding text are interesting measurements and model values that trigger some thoughts! First, at time 00-12 UTC 14 January Delta-Z-800hPa shows a clear pressure contrast, which then leads to a corresponding observed gap wind in the same time period. Hence, in this time slot Delta-Z-800hPa can be used as a reasonable diagnostic that quantifies the driving mechanism of the gap wind. However, then I also considered the time instance around 12 UTC 12 January. There, the diagnostic Delta-Z-800hPa is comparable in amplitude to the afore-mentioned time slot, but the observed wind in the valley is

completely different. Why is this? If my point is correct I wonder whether it is reasonable to assume that Delta-Z-800hPa can be taken as a metric for the gap-flow-driving mechanism? Possibly, I miss an essential point, or I misread the figure.

- Again to figure 3 and the corresponding text: It is first argued that the pressure contrast Delta-P-SFC cannot be used as a diagnostic for the gap-flow-driving mechanism. I fully agree, and it is also plausibly be argued why this field cannot be used. After having discussed that Delta-P-SFC cannot be used, the physically more reasonable Delta-Z-800hPa (at pass height) is introduced and discussed. The logic of the text structure assumes that the reader should expect that Delta-P-SFC should work as a diagnostic, but -- honestly -- it was clear to me from beginning that this won't work. Hence, the authors might start straightforward their discussion on the forcing mechanism with the much more plausible Delta-Z-800hPa.

- L173-174: 'The wind arrows... WRF simulation' -> Remove, no need to repeat this methodological aspect at this place.

- L174-175: 'Winds at Chamo ... at the pass' -> Describe in 1-2 sentences the differences in timing. Actually, this is done in the following sentence, but the 'Moreover' is a misleading connecting word between the two sentences.

- L188: correct 'tow' -> 'two'

- L194: It is stated that the temperature increase is consistent with the increase in pressure difference. I agree, but at the same time wonder what 'is consistent' exactly means. It this meant in a qualitative way, or is it assumed that the pressure contrast could be quantitatively be derived from the corresponding contrast in the temperature profiles, by applying the hydrostatic assumption?

- In Figure 5 and 6 the gap flow is from left to right, which is easy to 'read'. On the other hand, for the second case the flow in Figure 9 and 10 is from right to left. Of course, I see why the authors stick to this orientation. Personally, however, I would prefer an orientation of the vertical cross sections in a way that the flow also in Figure 9 and 10 goes from left to right. I think it would be easier to 'read'. However, I have no strong opinion on that, and only want the authors invite to re-think the orientation of the cross section in Figure 9 and 10.

- L196: 'a reduction of the flow depth...' -> How is the flow depth defined, how can it be inferred from the figures. Some more explanation would be helpful. It is, in the next sentence, also argued that this reduction indicates a transition to a supercritical state. I guess taht this is indeed the case, however, without quantitatively considering the Froude number the statement remains rather vague.

- Figure 10: 'Notice the reversed color scale compared to Fig. 6' -> That's OK, but if the orientation of the figures is reversed (flow from left to right, as mentioned before) this reversing of the color scale could be avoided.

- L334-335: Here it is argued that daytime convection may be a delaying mechanism, despite its role for differential heating. I think this specific role of convection is indeed a very interesting aspect. I would have appreciated a somewhat more detailed discussion how convection interacts/delays the gap flow. Actually, there are three effects discussed in the text that help to build up the temperature contrasts between the valley ends: (1) differential advection; (2) buildup (or not) of the CBL; and (3) the role of convection. Since this is a rather interesting aspect, a more careful discussion of the contributions and interactions would be nice.

- L343: 'the dynamic forcing was...' -> What is the dynamic forcing in this context? I guess it refers to the forcing due to the along-valley pressure gradient. However, due which degree is it reasonable to call this a dynamic forcer when the pressure contrast is driven by a temperature contrast? Possibly, I misunderstand the term 'dynamic forcing'?

- L343-344: ' Te thermal forcing.... on the smaller scale' -> I am not sure whether I fully understand the statement made here about larger- and smaller-scale effects. Please rephrase in clearer way.

- L363: Here, convective outflows are mentioned. Interesting! Is this the delaying effect that was earlier referred to?

 - L368-378: Here a comparison between WRF and ERA5 is made. This is, in principle, interesting and also relevant, since it tells that and to which degree ERA5 can be used for meteorological studies in the target region. Hence , there is no doubt that this should be included in the study. I wonder, however, if section 6 with its focus on physical mechanisms is the right place to do so? Personally, I would shift this aspect to the end of the conclusions, where it can be combined with the last bullet point and make a bridge to the final statement of the manuscript about future studies. Note also the (more methodological, data-based) ERA5/WRF topic interrupts the discussion on physical aspects before from the physical aspects after (on hydraulic theory).

- L382-384: 'Applying reduced-gravity.... Weiss, 2021)' -> Am I correct in assuming that these shallow-water experiments were indeed performed in Weiss (2021). This is not completely clear in superficially reading the text.

---

## Referee Comment (RC2)

Review for

Dynamics of Gap Winds in the Great Rift Valley, Ethiopia: Emphasis on Strong Winds at Lake Abaya
by Weiß et al

**Summary:**

The authors examined the structure of gap wind in the Great Rift Valley. In general, the introduction, methods, data, and almost of results are very clearly explained in this paper. However, there is a lack of explanation in some parts of the results and discussions, and the manuscript seems to require improvement. Therefore, I can recommend its publication in Weather and Climate Dynamics only after some major revisions.

**Major comments**

(1) Relation between $\Delta z_{800hPa}$ and Gap wind

The authors contend that $\Delta z_{800hPa}$ may be a predictor of gap wind in the target area. Indeed, $\Delta z$ and $\Delta p$ are the largest in the first half of the period when the Gap wind is strongest (Gray shade in Fig. 3b). However, in the second half of the period, $\Delta z$ (and $\Delta p$) are smaller. Furthermore, even during the period when the gap wind was not blowing, $\Delta z$ is comparable to those during the period when the gap wind was strong. Since the wind is driven by $\Delta z$ ($\Delta p$), they need to be linked. If gap wind did not blow despite the large $\Delta z$ ($\Delta p$), some local pressure gradient can be inhibiting gap wind. Therefore, the authors need to further elaborate on the structure of the atmosphere during the periods when the gap winds did not blow.

(2) Flow pattern of gap winds

The authors contend that thermally driven currents and pressure gradients (basin and plateau winds) play important roles in the flow pattern of the gap wind and its time evolution. However, a detailed analysis of the thermally driven flow is lacking. It is recommended to check not only $\Delta p$ and $\Delta z$ between the ends of the gap, but also $\Delta z$ and $\Delta p$ between the ends of the gap and inside the gap. Alternatively, numerical experiments without the ground surface heat flux might be helpful.

In addition, it seems very difficult for the readers to understand the flow pattern and time evolution of each of them only from the description in Section 6. Therefore, I recommend that the manuscript include conceptual flow diagrams for the northeast and southwest wind cases to assist the authors in their understanding.

(3) Hydraulic theory

   The authors contend that the localized strong winds at Lake Abaya in the southwesterly wind case were caused by a transition from subcritical to supercritical flow. What is the basis for this? It could simply be that the flow path has become narrower, and the wind velocity has increased. The authors should clearly show that the characteristics of the southwesterly wind cases are consistent with the qualitative characteristics of the transition flows.

**Specific comments**

(1) L92

evens -->events

(2) L111

domian --> domain

(3) L134

What is your basis for this assumption?

(4) L140

Why is it warm advection here?

(5) L143

What is responsible for this pressure gradient? Synoptic scale phenomena, temperature differences, or other things?

(6) L147

Why is the ITCZ not clearly visible from this figure?

(7) L150

Figure 2b shows that there is cold air advection from the southwest, but cold air advection from the Indian Ocean (southeast) does not appear to be present. Could you please explain exactly what you mean?

(8) L155

What do you mean by "the large-scale flow" here? On the flip side, did the large-scale flow not affect the gap winds in the first case? How did you determine the impact of the large-scale field from Figure 2?

(9) L170

Could you please indicate the time zone of the target area, e.g. in the introduction?

(10)L179

The authors need to explain why $\Delta p(\Delta z)$ does not match the gap wind strength.

(11) L188

  tow --> two

(12) L229

  I think we should break the line because the topic has changed.

(13) L231

  Are you referring to the local acceleration between GA and CN in Figure 6c? If so, it would be better to add something like "between GA and CN" to make it easier for the readers to identify the location.

(14) L289

  In what ways do you think this local acceleration resembles the transition from subcritical to supercritical flow? It could simply be that the flow path has become narrower, and the wind velocity has increased (Venturi effect).

  In addition, in the transition from supercritical flow to subcritical flow (the front of the Gap wind), a flow discontinuity such as a hydraulic jump is thought to occur. What is the cause of the lack of hydraulic jump at the head of this local acceleration region?

(15) L299

  Are these local accelerations caused by the same reason as the local accelerations over Lake Abaya?

(16) L321

  "distance of 800 km"

  Where is the distance between? I don't think there were any observation points as far apart as 800 km in your manuscript.

(17) L323

  Indeed, $\Delta z$ and $\Delta p$ are the largest in the first half of the period when the Gap wind is strongest (Gray shade in Fig. 3b). However, in the second half of the period, (and $\Delta p$) are smaller. Furthermore, even during the period when the gap wind was not blowing, $\Delta z$ is comparable to those during the period when the gap wind was strong. This hardly suggests that $\Delta z$ can be a useful predictor of gap winds in the target area.

  Since the wind is driven by $\Delta z$ ($\Delta p$), they need to be linked. If gap wind did not blow despite the large $\Delta z$ ($\Delta p$), some local pressure gradient can be inhibiting gap wind. However, Therefore, the authors need to further elaborate on the structure of the atmosphere during the periods when the gap

winds did not blow. Alternatively, I recommend deleting this sentence from the manuscript.

(18) Paragraph started from L345

It would be very difficult for readers to understand the pattern and time evolution of each gap wind event with only the description in section 6. Therefore, I propose that conceptual flow diagrams for the northeasterly and southwesterly wind cases be included in the manuscript to aid the reader's understanding.

(19)L360

If the basin winds are preventing the gap winds from penetrating the basin, there should be a flow into the basin. However, Figure 9a shows no such flow. Is there any reason why the Basin wind did not develop?

To answer this question, it is recommended to check not only $\Delta p$ and $\Delta z$ between the ends of the gap, but also $\Delta z$ and $\Delta p$ between the ends of the gap and inside the gap. Alternatively, numerical experiments without the ground surface heat flux might be helpful.

(20)L360

Do you mean that the gap wind near x=600 km in Figure 9a is localized because the gap wind in Figure 9c is blocked by the thermally driven flow? Could you describe it more clearly?

---

## Author Response (AR1)

Response to referees' comments

**Dynamics of Gap Winds in the Great Rift Valley, Ethiopia: Emphasis on Strong Winds at Lake Abaya**

DOI: 10.5194/wcd-2022-20

Cornelius I. Weiss, Alexander Gohm, Mathias W. Rotach Thomas T. Minda

June 26, 2022

**1. Introduction**

We thank both reviewers for their detailed feedback and their suggestions to improve the manuscript. In the following we provide a point-by-point answer to all reviewers' comments with the original comments in black and our responses in blue. In addition to the revised manuscript, we provide a version in which all changes have been highlighted in blue (i.e. parts added) and in red (i.e. parts removed)

**2. Response to referee 1**

**2.1. Synopsis**

The manuscripts present an interesting example of a gap wind in the Great Rift Valley, Ethopia, i.e., a region that attracted only little focus in atmospheric dynamics. Many dynamical mechanisms leading to the gap winds are discussed, e.g., the build-up of pressure difference along the value due to differential temperature advection, hydraulic flow behaviour due to valley slopes and constrictions, impact of convective activity,... I think the processes are well described, the figures of high quality, and the text is well structured and written in a clear, concise way. The only concern that could be brought up is that the evolution of the gap winds is discussed in a rather descriptive/qualitative way, without quantifying the forcing mechanisms. Still, it is a coherent story that is told about an interesting weather feature in a rarely studied region. Therefore, I can recommend publication of the study in *Weather and Climate Dynamics* if the following (mostly) minor comments are adequately addressed. In addition to physical aspects, I also make a few suggestions about wording.

Thank you for this very positive feedback. We are aware that the manuscript is of a rather descriptive nature. This is the first study describing the gap winds in the Great Rift Valley (GRV) in Ethiopia, and our main aim was to raise awareness of this phenomenon and suggest potential drivers. Nevertheless, we tried to quantify some of the main drivers like the large-scale pressure gradient (e.g. Fig. 3a and Fig. 7a). To further separate local, regional, and large-scale drivers would require a more systematic numerical sensitivity study, which is beyond the scope of the present study. However, we hope that this paper triggers subsequent research in this direction since the phenomenon is of great importance for the local population.

**2.2. Comments**

-L4: 'Therefore, two...' -> To this aim, two...'
We changed that.
-L15: 'were location dependent' -> depended on location'

We changed that.

-L17,18: 'The ERA5... Hence, ERA5...gap flows': I am not sure weather keep this more data-related aspect in this more mechanism-focused abstract.

We kept it in the abstract, especially because it adds important information beyond the the physical process and provides a starting point for future studies, such as a climatology of gap winds in the GRV based on reanalysis data.

- L20: 'produce water waves' -> 'induce surface waves on the lake'

We changed that.

- L29: 'The average depth of the lake is 8.6 m' -> It is interesting, although beyond the scope of this study, to think about how this rather shallow lake depth affects the waves that form on the surface. Possibly, the authors want to refer to one, two studies that address the modeling of surface waves on lakes, e.g.,

*Seibt, Christian, Peeters, Frank, Graf, Michael, Sprenger, Michael, Hofmann, Hilmar, (2012), Modeling wind waves and wave exposure of nearshore zones in medium-sized lakes, Limnology and Oceanography, 58, doi: 10.4319/lo.2013.58.1.0023.*

*Graf, Michael, et al. "Evaluating the suitability of the SWAN/COSMO-2 modelsystem to simulate short-crested surface waves for a narrow lake with complex bathymetry." Meteorologische Zeitschrift 22.3 (2013): 257-272.*

The two studies illustrate how the ecology, but also archeological sites are affected by waves, in particular in the shallow shore region. I do not think that this wave-on-the-lake aspect is crucial for the study, but it is a nice point how strong winds can have indirect effects/impacts. If the introduction could be shortened, it would certainly be this relatively long paragraph (L27-50) about ecology and waves. However, as mentiond before, this is not mandatory from my side.

Aspect 1) Shallow lake depth: That is a rather interesting point and it could be very advantageous to couple a lake model like SWAN (Seibt et al. (2012) and Graf et al. (2013)) to an atmospheric model to investigate the wave development on Lake Abaya. However, again, this is beyond the scope of this study.

Aspect 2) Wind impact: We kept the wave-on-the-lake aspect paragraph in the introduction, since it shortly describes the motivation for our study and the relevance of the gap winds for the local population. For the explanation of the gap winds itself, however, its not relevant.

- L45: 'a bidirectional effect' -> 'two effects'

We changed that.

- L55,56: Avoid two consecutive sentences starting with similar 'connecting' words (Hence, Therefore)

We changed that by removing 'therefore' at the beginning of the second sentence.

- L52: 'Besides other large-scale drivers like ENSO...' -> How relevant is ENSO for *weather* in Ethopia? I am not sure that such a climate index is particularly relevant for this study. If yes, it could be discussed in somewhat greater detail

ENSO affects the weather, especially precipitation (which is linked to atmospheric motion) in Ethiopia (Gleixner et al., 2017). For this study, however, the position of the ITCZ is of main importance, hence, we removed the ENSO part of the sentence.

- L58: 'dual passage' -> 'biannual | twice-yearly passage'

We changed that.

- L69: In L49-50 it is written that the the waves could be the reason why Lake Abaya is less affected by invasive plants than, e.g., Lake Tana. In L45 it is argued that it is Lake Abaya's large size that allows many waves to evolve because of the large wind tech. On the other hand, Lake Tana is larger, offering a larger wind fetch. Wouldn't one expect larger waves? Hence, is it more the wind speed that determines the wave activity than the wind fetch

We have to admit that Lake Tana is not the best comparison here, since the lake is not within the GRV. However, Lake Abaye is the largest lake in the Ethiopian GRV and is less affected by invasive plants. We suppose that this is caused by larger waves (assumed that all GRV lakes experience comparable wind speeds). In comparison to Lake Tana, wind speed might play a more important role than the size of the lake, thats true. So we changed that sentence to 'than other lakes in the Ethiopian GRV.'.

- L63: 'The adjustment of the wind field by local effects' -> What does this exactly mean? Could you be somewhat more specific!

We changed the wording: The sentence now reads: "These mesoscale winds strongly vary on the daily and seasonal scale".

- L66: 'shows strong differences to the synoptic flow' -> 'strongly deviates from the synoptic flow'

We changed that.

- L69: 'Lake Abaya is at least in some months connected to the large scale flow' -> Could you be a little more specific? What does 'connect' mean in this context?

We changed it to: '.... that in some months the local winds around Lake Abaya blow in the same direction as the synoptic flow. Hence, forced channeling of the large-scale flow may influence or even determine the local flow (e.g. Kossmann and Sturman, 2003).' We also changed the second sentence to: '... the impact of the synoptic flow is strongest in May and June...'

- L73: 'on the lake water dynamics' -> 'on the lake's wave dynamics and internal mixing'

We changed that.

- L74: 'Therefore these winds...' -> 'It is the aim of this study to better understand these winds...'

We changed that.

- L76: 'the goal of' -> 'the more specific goal of'

We changed that.

- L79: 'The synoptic conditions prior to the two cases' -> Why *prior* to these events?

We removed 'prior'.

- L89: Remove 'that allowed for the local perspective'

We removed 'that allowed for the local perspective'

- L92: 'horizontal resolution of 0.25 x 0.25' -> make clear that ERA5 is based on a spectral model and thus provide the spectral resolution, the grid spacing in latitude/longitude is 'only' an interpolation from spectral space

We agree that ERA5 is based on a spectral model. However, this information is not of great importance for the reader. We think it is more important to mention the resolution of the data set, which is distributed via the Coperincus Climate Data Store (CDS) and which we had access to. However, we added the information to the data source (CDS) where we obtained the data.

- L101: 'ERA5 model topography. ERA5 topography near...' -> 'ERA5 model topography, which near ...'

We changed that.

- L140: 'Lake Abaya, hence, the at the...' -> correct sentence structure!

We changed that to 'Lake Abaye, i.e. at the'.

- L141-143: 'led to an along pressure' -> more precisely, 'led to an along pressure gradient'

We changed that.

- L142: 'height Z at the 800 hPa surface. 800 hPa...' -> 'height Z at the 800 hPa surface, which...'

We changed that.

- L143: Here it is written that the pressure gradient *helped* to channel the air though the gap. The term 'helped' is somewhat to o unspecific? Which other processes drive the air through the channel? To which degree does the pressure gradient contributes to the flow through the valley, and to which degree are other processes (which?) essential? I am quite convinced that the pressure gradient is indeed decisive, but the wording in the text could be more careful. In the same line of argument I wonder to which degree it is possible to understand the pressure gradient at 800 hPa by means of hydrostatic effects, i.e., due to the differences in the teperature?

We split the answer into three parts:

1) Wording 'helped': We changed it to 'provided the basis' as we think that the pressure gradient played the major role in the evolution of the gap flow.

2) Other processes: Processes besides the pressure gradient were crucial for the development of the gap glow in our opinion. The diurnal cycle of the temperature gradient and, hence, the diurnal cycle of the pressure gradient was important too. However, this is not visible in Fig. 2 but in the consecutive figures, especially Fig. 3 and 4. One can see this diurnal cycle also by comparing synoptic maps for different times.

However, since we only show one synoptic map in our manuscript, we do not mention other processes at this point. Uncovering information piece by piece is our way of storytelling. Moreover, at this stage we cannot quantify to which degree the large-scale and local pressure gradients determined the gap flow. A deeper investigation based on sensitivity studies would be necessary.

3) Hydrostatic effects: Since the ERA5 data is based on an hydrostatic model, the pressure gradients depicted by the reanalysis are of purely hydrostatic nature. This can also be seen from the strong relation between pressure and temperature gradients illustrated by the vertical profiles in FIg. 4 and 8.

- L155-157: Here, it is state that the large-scale flow is non-negligible, because it caused an inflow and hence forced channeling at both valley entrances. I am not sure whether I understand this point, and whether I see this inflow channeling in the figure. Possibly, my point is also related to the following question: Is the large-scale flow also associated with a pressure gradient that is perpendicular to the valley axis and thus contributes to the along-valley acceleration of the flow. In short, a more careful discussion/distinction of local effects and large-scale effects would be helpful.

We split the answer into two parts:

1) Large-scale inflow: At both valley ends, the large scale winds associated with the position of the ITCZ (weakly) penetrate into the GRV. This is the forced channeling we refer to. In the paper, we tried to improve the description of the large-scale effects.

2) Pressure gradient: The large-scale pressure gradient was pointing in along-valley direction, indicated by the isohypses (blue lines) in Fig. 2. Notice that the large-scale winds at low latitudes are generally not in geostrophic balance since the Coriolis force is weak and, hence, do not blow parallel to the isohypses on pressure levels. We added this remark in the manuscript. We also clarified the difference between pressure-driven channeling and forced channeling by referring to Kossmann and Sturman (2003).

- L162.163: The pressure difference...and their mixing' -> I had to read this sentence several times, and still I am not sure that I get it correctly. Please rephrase in clearer way. Possibly, a way to do so would be: 'Two points are selected at the valley entrance and exit, and the pressure difference between these two points is then take as a diagnostic for the gap winds in the GRV and their temporal evolution'.

We changed that to 'Two ERA5 grid points are selected at the two valley entrances, and the pressure difference between these two points is then taken as a diagnostic to describe the temporal evolution of the gap winds in the GRV.'

- Figure 3 and the corresponding text are interesting measurements and model values that trigger some thoughts! First, at time 00-12 UTC 14 January Delta-Z-800hPa shows a clear pressure contrast, which then leads to a corresponding observed gap wind in the same time period. Hence, in this time slot Delta-Z-800hPa can be used as a reasonable diagnostic that quantifies the driving mechanism of the gap wind. However, then I also considered the time instance around 12 UTC 12 January. There, the diagnostic Delta-Z-800hPa is comparable in amplitude to the afore-mentioned time slot, but the observed wind in the valley is completely different. Why is this? If my point is correct I wonder whether it is reasonable to assume that Delta-Z-800hPa can be taken as a metric for the gap-flow-driving mechanism? Possibly, I miss an essential point, or I misread the figure.

We understand you concern but still think that $\Delta Z$ at 800 hPa is a reasonable diagnostic for gap winds. Part of the problem comes from the fact that the large-scale signal of the pressure gradient is strongly modulated by the diurnal cycle. The daily mean values in $\Delta Z$ of the two time slots you mentioned differ by about 10 m. In Fig. R1 we show a 24-h running mean of $\Delta Z$ in which the diurnal cycle has been filtered. It is clearly visible that the mean value of $\Delta Z$ is highest during the time of the gap wind. A threshold of about $\Delta Z=15$ m at 800 hPa needs to be exceeded to produce gap winds. A peak in the filtered $\Delta Z$ in the period of gap winds can also be seen at 700 hPa, but less pronounced. We added the filtered signal also in the manuscript.

- Again to figure 3 and the corresponding text: It is first argued that the pressure contrast Delta-P-SFC cannot be used as a diagnostic for the gap-flow-driving mechanism. I fully agree, and it is also plausibly be argued why this field cannot be used. After having discussed that Delta-P-SFC cannot be used, the physically more reasonable Delta-Z-800hPa (at pass height) is introduced and discussed. The logic of the text structure assumes that the reader should expect that Delta-P-SFC should work as a diagnostic,

[Figure]

Figure R1: As Fig. 3 in the manuscript but with a 24 h running mean for ΔZ to emphasis the increase in the pressure gradient at pass height over the course of the days.

but – honestly – it was clear to me from beginning that this won't work. Hence, the authors might start straightforward their discussion on the forcing mechanism with the much more plausible Delta-Z-800hPa.

Thanks for this hint. We started with $\Delta P_{sfc}$ because it is a measure, that is widely used in predicting *foehn* in the European Alps such as at Innsbruck (e.g. `https://www.windinfo.eu/wettervorhersage/foehndiagramme/`). Obviously, this diagnostic is also not the best choice for Alpine locations. We kept it in the manuscript to give a hint for future studies not using the surface pressure difference.

- L173-174: 'The wind arrows... WRF simulation' -> Remove, no need to repeat this methodological aspect at this place.

We removed that.

- L174-175: 'Winds at Chamo ... at the pass' -> Describe in 1-2 sentences the differences in timing. Actually, this is done in the following sentence, but the 'Moreover' is a misleading connecting word between the two sentences.

We changed that to ' Winds at Chamo North were generally stronger and occurred later than at the pass. The strongest winds...'

- L188: correct 'tow' -> 'two'

We changed that.

- L194: It is stated that the temperature increase is consistent with the increase in pressure difference. I agree, but at the same time wonder what 'is consistent' exactly means. It this meant in a qualitative way, or is it assumed that the pressure contrast could be quantitatively be derived from the corresponding contrast in the temperature profiles, by applying the hydrostatic assumption?

Yes, the pressure gradient derived from hydrostatic integration of the two temperature profiles should be essentially the same as the pressure gradient calculated directly from the two pressure profiles (we did not proof). For ERA5 this has to be the case since it is based on a hydrostatic model. But also for the WRF

model the hydrostatic part must be dominant one as the good agreement between WRF and ERA5 shows.

- In Figure 5 and 6 the gap flow is from left to right, which is easy to 'read'. On the other hand, for the second case the flow in Figure 9 and 10 is from right to left. Of course, I see why the authors stick to this orientation. Personally, however, I would prefer an orientation of the vertical cross sections in a way that the flow also in Figure 9 and 10 goes from left to right. I think it would be easier to 'read'. However, I have no strong opinion on that, and only want the authors invite to re-think the orientation of the cross section in Figure 9 and 10.

We agree that vertical cross sections are usually oriented to show the prevailing winds from left to right. This applies for the first event. However, changing the orientation of the transect of the second event just to stick to this "silent agreement" would probably even more confuse the reader. Hence, we decided to use the same transect alignment for both events.

- L196: 'a reduction of the flow depth...' -> How is the flow depth defined, how can it be inferred from the figures. Some more explanation would be helpful. It is, in the next sentence, also argued that this reduction indicates a transition to a supercritical state. I guess taht this is indeed the case, however, without quantitatively considering the Froude number the statement remains rather vague.

L196 in the reviewers' comment should probably refer to L296. We did not include a Froude number analysis in the manuscript since it is non-trivial for a continuously stratified fluid as shown by, e.g., Weiß (2021). Due to the ambiguity of determining, e.g., the reduced gravity the comparison between hydraulic flow and atmospheric flow should rather be done in a qualitative sense (e.g., Durran, 2015). However, we believe that there is no doubt that the descending isentropes mentioned in the text resemble a transition into a supercritical state. We added the following remark to the manuscript: "The acceleration was associated with descending isentropes and, hence, a reduction of the flow depth (see layer below the 316-K isentrope which also marks the height of the flow reversal)."

- Figure 10: 'Notice the reversed color scale compared to Fig. 6' -> That's OK, but if the orientation of the figures is reversed (flow from left to right, as mentioned before) this reversing of the color scale could be avoided.

See comment to Figure 5 and 6 above.

- L334-335: Here it is argued that daytime convection may be a delaying mechanism, despite its role for differential heating. I think this specific role of convection is indeed a very interesting aspect. I would have appreciated a somewhat more detailed discussion how convection interacts/delays the gap flow. Actually, there are three effects discussed in the text that help to build up the temperature contrasts between the valley ends: (1) differential advection; (2) buildup (or not) of the CBL; and (3) the role of convection. Since this is a rather interesting aspect, a more careful discussion of the contributions and interactions would be nice.

We agree that convection is an interesting aspect whose role should be clarified in a future sensitivity study. Clearly, this is beyond the scope of the present study, and so we can do no more than mention a possible connection between the decaying convection and the onset of the gap flow. Hopefully, this qualitative statement provides motivation for a future study.

- L343: 'the dynamic forcing was...' -> What is the dynamic forcing in this context? I guess it refers to the forcing due to the along-valley pressure gradient. However, due which degree is it reasonable to call this a dynamic forcer when the pressure contrast is driven by a temperature contrast? Possibly, I misunderstand the term 'dynamic forcing'?

We changed that to '...cases the synoptic forcing due to an air mass difference across the GRV was not able... '.

- L343-344: ' Te thermal forcing.... on the smaller scale' -> I am not sure whether I fully understand the statement made here about l arger- and smaller-scale effects. Please rephrase in clearer way.

We changed that to 'The diurnal forcing was an important supporter, if not a key player, on the larger scale by enhancing the air mass difference due to a different CBL growth but presumably a delayer on the smaller scale due to counteracting valley winds and convection on the lee slope.'

- L363: Here, convective outflows are mentioned. Interesting! Is this the delaying effect that was earlier referred to?

Yes, it could have caused a delaying effect. Further research is necessary to clarify the effect of convection on different scales.

- L368-378: Here a comparison between WRF and ERA5 is made. This is, in principle, interesting and also relevant, since it tells that and to which degree ERA5 can be used for meteorological studies in the target region. Hence , there is no doubt that this should be included in the study. I wonder, however, if section 6 with its focus on physical mechanisms is the right place to do so? Personally, I would shift this aspect to the end of the conclusions, where it can be combined with the last bullet point and make a bridge to the final statement of the manuscript about future studies. Note also the (more methodological, data-based) ERA5/WRF topic interrupts the discussion on physical aspects before from the physical aspects after (on hydraulic theory).

We would like to keep it in the discussion (to keep the conclusion as short as possible), but we shifted it behind the passage about hydraulic theory to not interrupt the more physically based aspects of the discussion. Thanks for this suggestion.

- L382-384: 'Applying reduced-gravity.... Weiss, 2021' -> Am I correct in assuming that these shallow-water experiments were indeed performed in Weiss (2021). This is not completely clear in superficially reading the text.

Yes, that is correct. We rephrased it to 'Applying reduced-gravity shallow-water theory (see Weiß, 2021) supported the expected transition...'. However, it is noteworthy that we did not run a hydraulic model in Weiß (2021) but rather estimated the Froude number, the non-dimensional mountain height and the channel width and derived the resulting flow regime from a regime diagram.

**3. RESPONSE TO REFEREE 2**

**3.1. Summary**

The authors examined the structure of gap wind in the Great Rift Valley. In general, the introduction, methods, data, and almost of results are very clearly explained in this paper. However, there is a lack of explanation in some parts of the results and discussions, and the manuscript seems to require improvement. Therefore, I can recommend its publication in Weather and Climate Dynamics only after some major revisions.

We appreciate your feedback and tried to improve our manuscript.

**3.2. Major comments**

(1) Relation between Δz 800hPa and Gap wind The authors contend that Δz 800hPa may be a predictor of gap wind in the target area. Indeed, Δz and Δp are the largest in the first half of the period when the Gap wind is strongest (Gray shade in Fig. 3b). However, in the second half of the period, Δz (and Δp) are smaller. Furthermore, even during the period when the gap wind was not blowing, Δz is comparable to those during the period when the gap wind was strong. Since the wind is driven by Δz (Δp), they need to be linked. If gap wind did not blow despite the large Δz (Δp), some local pressure gradient can be inhibiting gap wind. Therefore, the authors need to further elaborate on the structure of the atmosphere during the periods when the gap winds did not blow.

We split the answer in two part:

1) Lower ΔZ in second half of the period: It is true, that the signal of ΔZ at both, 800 and 700 hPa, reached a maximum in the first half of the marked period. This reflects the enhancement of the air mass difference by the differential growth of the CBL during the day. We have to keep in mind, however, the Great Rift Valley is a few hundred km long. This implies, that we can expect a time lag due to a phase shift between the largest along-valley pressure gradient and the resulting local winds on the downstream side of the gap. The occurrence of the strongest gap winds within the valley is therefore location dependent (see, e.g., differences between ERA5 winds at the pass and the observed and simulated winds at Chamo North in Fig. 3c). Moreover, when filtering the diurnal cycle by calculating a 24-h running mean of ΔZ (shown in Fig. R1b in this document and now also in the manuscript), we see that the smoothed signal is very similar in

both halves of the shaded period.

2) High $\Delta Z$ without strong gap winds: we assume, that the reviewer is referring to one of the days before the considered gap flow event (e.g., 12 UTC on 12 and 13 January 2020). Indeed the wind should be linked to the pressure gradient. Looking at the 24-h running mean (Fig. R1b), we see that $\Delta Z$ was approximately 10 m lower at the days before the investigated event. Hence, it appears that a threshold of about $\Delta Z$=15 m at 800 hPa has to be exceeded for gap flows to occur.. Additionally, the daily cycle was less pronounced at these days. Nevertheless, in the night from 13 to 14 January, when the pressure gradient already increased, a weaker gap flow event was initiated. This supports our idea that $\Delta Z$ can be linked to the occurrence and strength of the gap winds in the GRV.

(2) Flow pattern of gap winds The authors contend that thermally driven currents and pressure gradients (basin and plateau winds) play important roles in the flow pattern of the gap wind and its time evolution. However, a detailed analysis of the thermally driven flow is lacking. It is recommended to check not only $\Delta p$ and $\Delta z$ between the ends of the gap, but also $\Delta z$ and $\Delta p$ between the ends of the gap and inside the gap. Alternatively, numerical experiments without the ground surface heat flux might be helpful. In addition, it seems very difficult for the readers to understand the flow pattern and time evolution of each of them only from the description in Section 6. Therefore, I recommend that the manuscript include conceptual flow diagrams for the northeast and southwest wind cases to assist the authors in their understanding.

We split the answer into three parts:

1) Thermally driven flows: We agree, that both phenomena had a strong thermal forcing component, however, we are not able to provide a detailed quantification of the single components based on our simulations. Therefore, as suggested, additional numerical sensitivity experiments would be necessary, e.g., changing the magnitude of the surface sensible heat flux or modifying the topography. Unfortunately, this is beyond the scope of the present study, but we hope to provide a starting point for future research concerning gap winds in the GRV. We tried to describe the different thermal contributors to the best of our knowledge.

2) Concerning the recommendation about pressure gradient between the end of the gap and inside the gap: This is already given in the vertical profiles of the pressure gradient in Fig. 4 and 8. Lake Abaya is inside the valley and Aledeghi at the northeastern end (cf. Fig.1b in the manuscript).

3) Conceptual flow diagram: Thanks for this valuable recommendation. We thought about that but did not find a satisfying solution. Since many processes on different time and spatial scales are involved, it is not trivial. Instead of a schematic diagram, we tried to write a concise conclusion, where the main statements are divided into short paragraphs.

(3) Hydraulic theory The authors contend that the localized strong winds at Lake Abaya in the southwesterly wind case were caused by a transition from subcritical to supercritical flow. What is the basis for this? It could simply be that the flow path has become narrower, and the wind velocity has increased. The authors should clearly show that the characteristics of the southwesterly wind cases are consistent with the qualitative characteristics of the transition flows.

We assume, that the comment refers to Fig. 9a or 10b in the manuscript. Looking at Fig. 1c, one can see that the local valley width is narrowest at the middle of Lake Abaya (point AG in Fig. 10b), hence, the flow has to converge which accelerates the flow. The strongest winds, however, are downstream of the narrowest section. This is a key feature of a gap flow and a hydraulic transition of a sub- to a supercritical flow. Jackson et al. (2013), e.g., state that "the maximum [wind speed] is close to the exit of the gap and not at its narrowest part as simple continuity reasoning ("Venturi flow") might suggest". The asymmetric structure is usually explained by a hydraulic response, i.e., a transition from a sub- to a supercritical state (sec. 3 Arakawa, 1969). Hence, we interpret this flow acceleration as a hydraulic response rather than as a *Venturi flow*. In the manuscript we added in L288 (old) the following sentence to specify our argument: 'The strongest winds occurred downstream of the narrowest section of the valley, i.e., downstream of the point AG. This is similar to a hydraulic flow transition...'

**3.3. Specific comments**

(1) L92 evens –>events

We changed that.

(2) L111 domian –> domain

We changed that.

(3) L134 What is your basis for this assumption?

We have not tested whether deviations of 1 to 2 K in lake temperature impacts the simulated flow structure. But it seems to be pretty clear that the effect of such small deviations is negligible given the fact that land and see breezes are driven by temperature contrasts being about one order of magnitude larger. Nevertheless, we removed the sentence "This difference was assumed to have no significant influence on the simulation results".

(4) L140 Why is it warm advection here?

The warm air advection happened a few days before. We changed it to '...potentially warmer air was present. This different air masses led to an along-valley pressure gradient ...'

(5) L143 What is responsible for this pressure gradient? Synoptic scale phenomena, temperature differences, or other things?

As described in the sentence before, the pressure gradient was established by the different air masses, hence, we would refer to as a synoptic scale phenomena. The enhancement of the pressure gradient by the different CBL growth is mentioned later when visible in Fig. 4.

(6) L147 Why is the ITCZ not clearly visible from this figure?

The ITCZ is not clearly visible because it is strongly fragmented by and deformed by the Ethiopian Highlands. On a larger chart snippet it is easier to identify. However, one can see it on the right border of the left panel of Fig. 2 in the manuscript where the wind arrows converge (around $9.5°$). This is along the Ethiopian Highlands, which are lower than 2000 m in this area. At the northwestern edge of the Ethiopian Highlands (not in Fig. 2 anymore), the ITCZ is better visible again.

(7) L150 Figure 2b shows that there is cold air advection from the southwest, but cold air advection from the Indian Ocean (southeast) does not appear to be present. Could you please explain exactly what you mean?

The air originates from the southern Indian Ocean and is than deflected by the massive of Mt. Kenya towards the northeast, resulting in the southeasterly winds on Fig. 2b. A branch of this southeasterly air stream is deflected into the Turkana Channel.

(8) L155 What do you mean by "the large-scale flow" here? On the flip side, did the large-scale flow not affect the gap winds in the first case? How did you determine the impact of the large-scale field from Figure 2?

We split the answer into two parts:

1) Large-scale inflow: At both valley ends, the large scale winds associated with the ITCZ (weakly) penetrate into the GRV. This is the forced channeling we refer to. In the paper, we tried to improve the description of the large-scale effects.

2) Large-scale flow for northeast case: For this case, no inflow into the GRV from the southwest was seen, that could be associated with the wind field in the Turkana Channel.

(9) L170 Could you please indicate the time zone of the target area, e.g. in the introduction?

We introduce the time zone at line 175 (i.e., five lines further below) when we talk the first time about a specific time.

(10)L179 The authors need to explain why Δp(Δz) does not match the gap wind strength.

We assume that this comment refers to the Δp at the surface. Δp at the surface (i.e. below pass height) is strongly determined by the boundary layer evolution below pass height. It is therefore a bad proxy for the pressure gradient at pass height. Hence, ΔZ near pass height (800 hPa) is more suitable to explain the timing and strength of the gap winds (see Sec. 3.2 point 1).

(11) L188 tow –> two

We changed that.

(12) L229 I think we should break the line because the topic has changed.

We changed that.

(13) L231 Are you referring to the local acceleration between GA and CN in Figure 6c? If so, it would

be better to add something like "between GA and CN" to make it easier for the readers to identify the location.

We changed that.

(14) L289 In what ways do you think this local acceleration resembles the transition from subcritical to supercritical flow? It could simply be that the flow path has become narrower, and the wind velocity has increased (Venturi effect). In addition, in the transition from supercritical flow to subcritical flow (the front of the Gap wind), a flow discontinuity such as a hydraulic jump is thought to occur. What is the cause of the lack of hydraulic jump at the head of this local acceleration region?

We split the answer into two parts:

1) Hydraulic response: see Sec. 3.2 point (3)

2) Missing hydraulic jump: In Fig. 10b) in the manuscript we see a retransition of the flow depth to the initial level beginning above LA. It is, however, not perfectly distinct since the flow approaches already the next obstacle, i.e. the pass.

(15) L299 Are these local accelerations caused by the same reason as the local accelerations over Lake Abaya?

Yes, the flow accelerations are similar to the acceleration over Lake Abaya due to a hill between Lake Chamo and Lake Abaya and the local narrowing, respectively. In the manuscript we added '...similar to the accelerations at Lake Abaya.'

(16) L321 "distance of 800 km" Where is the distance between? I don't think there were any observation points as far apart as 800 km in your manuscript.

The direct distance between the two selected ERA5 grid points is about 750 km. Following the valley axis it is around 800 km (see distance on the x-axis of the cross sections). We changed it to 'more than 700 km'.

(17) L323 Indeed, $\Delta z$ and $\Delta p$ are the largest in the first half of the period when the Gap wind is strongest (Gray shade in Fig. 3b). However, in the second half of the period, (and $\Delta p$) are smaller. Furthermore, even during the period when the gap wind was not blowing, $\Delta z$ is comparable to those during the period when the gap wind was strong. This hardly suggests that $\Delta z$ can be a useful predictor of gap winds in the target area. Since the wind is driven by $\Delta z$ ($\Delta p$), they need to be linked. If gap wind did not blow despite the large $\Delta z$ ($\Delta p$), some local pressure gradient can be inhibiting gap wind. However, Therefore, the authors need to further elaborate on the structure of the atmosphere during the periods when the gapwinds did not blow. Alternatively, I recommend deleting this sentence from the manuscript.

See Sec. 3.2 point (1). Moreover, we kept the sentence about $\Delta Z$ at 800 hPa being a possible predictor but we could imagine that there exists an even better predictor that incorporates other parameters too.

(18) Paragraph started from L345 It would be very difficult for readers to understand the pattern and time evolution of each gap wind event with only the description in section 6. Therefore, I propose that conceptual flow diagrams for the northeasterly and southwesterly wind cases be included in the manuscript to aid the reader's understanding.

See Sec. 3.2 point (2).

(19)L360 If the basin winds are preventing the gap winds from penetrating the basin, there should be a flow into the basin. However, Figure 9a shows no such flow. Is there any reason why the Basin wind did not develop? To answer this question, it is recommended to check not only $\Delta p$ and $\Delta z$ between the ends of the gap, but also $\Delta z$ and $\Delta p$ between the ends of the gap and inside the gap. Alternatively, numerical experiments without the ground surface heat flux might be helpful.

We split the answer into two parts:

1) Basin winds: Since the reviewer refers to Fig. 9a, we focus on the southwest case. We did not mention basin winds (just plateau winds) with respect to the southwest case since we did not feel confident to talk about them. Nevertheless, you are right that one could expect basin winds from the northeast penetrating into the GRV until the pass. As a consequence, in Fig. 9a there should be a flow from AL to PA which is not depicted within the valley. Without sensitivity experiments we have no answer, why the basin winds did not establish in this case. We can only assume that the weak southwesterly winds were stronger than possible basin winds.

2) ΔZ distance and sensitivity experiments: see Sec. 3.2 point (2).

(20)L360 Do you mean that the gap wind near x=600 km in Figure 9a is localized because the gap wind in Figure 9c is blocked by the thermally driven flow? Could you describe it more clearly?

We do not understand where the gap wind is blocked in Fig. 9c, hence, we do not know how to respond. Do you mean blocked upstream of the pass? Probably there is a typo in the line number, however, we are not able to find the corresponding part in the manuscript.

**REFERENCES**

Arakawa, S.: Climatological and dynamical studies on the local strong winds, mainly in Hokkaido, Japan, Geophysical magazine, 34, 359–425, 1969.

Durran, D.: MOUNTAIN METEOROLOGY | Downslope Winds, in: Encyclopedia of Atmospheric Sciences (Second Edition), edited by North, G. R., Pyle, J., and Zhang, F., pp. 69–74, Academic Press, Oxford, second edition edn., https://doi.org/https://doi.org/10.1016/B978-0-12-382225-3.00288-7, 2015.

Gleixner, S., Keenlyside, N., Viste, E., and Korecha, D.: The El Niño effect on Ethiopian summer rainfall, Climate Dynamics, 49, 1865–1883, https://doi.org/10.1007/s00382-016-3421-z, 2017.

Jackson, P. L., Mayr, G. J., and Vosper, S.: Dynamically-Driven Winds, https://doi.org/10.1007/978-94-007-4098-3_3, 2013.

Kossmann, M. and Sturman, A. P.: Pressure-driven channeling effects in bent valleys, Journal of Applied Meteorology, 42, 151–158, https://doi.org/10.1175/1520-0450(2003)042<0151:PDCEIB>2.0.CO;2, 2003.

Weiß, C.: Dynamics of Gap Winds in the Great Rift Valley, Ethiopia: Emphasis on Strong Winds at Lake Abaya, Master's thesis, University of Innsbruck, URL `https://resolver.obvsg.at/urn:nbn:at:at-ubi:1-96956`, (last access: 25 June 2022), 2021.

---

## Referee Report (RR1)

Review for

Dynamics of Gap Winds in the Great Rift Valley, Ethiopia: Emphasis on Strong Winds at Lake Abaya by Weiß et al

**Summary:**

Thank you for revising the manuscript. The authors have answered my questions adequately. However, the authors have not revised the manuscript based on some responses. I hope the authors not only answer the questions but also revise their manuscript. If they do not revise, I hope they will also write the reasons for not doing so. For the above reasons, I can recommend its publication in Weather and Climate Dynamics only after some minor revisions.

-Response to RC on Major comment (2) of reviewer 2.

I agree with the authors' explanation. However, if the authors argue that $\Delta z$ is an important factor in predicting gap winds in GRV, I think you should provide quantitative values for when the gap winds occur, even if it is just one example. So, I think that it is necessary to write somewhere in the manuscript the information that the gap wind occur when the $\Delta Z$ exceeded 15 m.

-Response to RC on 3.3. (7) of reviewer 2.

I agree with the authors' explanation. However, this information does not seem to be present in the manuscript. I think this information needs to be included in the manuscript as well for the benefit of readers who are not familiar with this area.

-Response to RC on 3.3. (16) of reviewer 2.

I understand. However, while the specific numbers of the distances are of course important, it is necessary to write "between where" in the manuscript so that "the readers can understand".

- Response to RC on 3.3. (19)-(2) of reviewer 2 (Just a comment, no need to reflect the manuscript).

I am sorry that I could not accurately inform you of my intentions. In the evening (14 UTC, Fig. 9a), gap winds develop near AG (between x=550 and x = 650 km). On the other hand, at night (19UTC, Fig. 9c), the gap winds near AG are weak and gap winds develop between x=100 and x=500 km. In other words, in the evening (14UTC), the location of the gap winds is limited to the vicinity of AG due to the thermally localized circulation (pressure gradient that drives the plateau and basin winds), and at night (19UTC), the gap winds occur over the entire valley because the thermally localized circulation decays. Is it correct? The authors think that this is probably future studies. I am looking forward to it.

---

## Author Response (AR2)

Second response to reviewer's and co-editor's comments

**Dynamics of Gap Winds in the Great Rift Valley, Ethiopia: Emphasis on Strong Winds at Lake Abaya**

DOI: 10.5194/wcd-2022-20

Cornelius I. Weiss, Alexander Gohm, Mathias W. Rotach Thomas T. Minda

July 19, 2022

**1. Introduction**

We thank the reviewer and the co-editor for their second feedback and the suggestions to improve the manuscript. In the following we provide a point-by-point answer to their comments with the original comments in black and our responses in blue. In addition to the revised manuscript, we provide a version in which all changes have been highlighted in blue (i.e. parts added) and in red (i.e. parts removed)

**2. Response to referee 2**

**2.1. Synopsis**

Thank you for revising the manuscript. The authors have answered my questions adequately. However, the authors have not revised the manuscript based on some responses. I hope the authors not only answer the questions but also revise their manuscript. If they do not revise, I hope they will also write the reasons for not doing so. For the above reasons, I can recommend its publication in Weather and Climate Dynamics only after some minor revisions.

We are sorry that our previous revision was not complete enough and hope that the new changes have now sufficiently addressed the remaining concerns.

**2.2. Comments**

-Response to RC on Major comment (2) of reviewer 2.
I agree with the authors' explanation. However, if the authors argue that $\Delta Z$ is an important factor in predicting gap winds in GRV, I think you should provide quantitative values for when the gap winds occur, even if it is just one example. So, I think that it is necessary to write somewhere in the manuscript the information that the gap wind occur when the $\Delta Z$ exceeded 15 m

We agree and changed that. In the manuscript (L190) it reads now: '... Fig. 3b shows a 24-hour running mean of $\Delta Z$ in which the diurnal cycle has been filtered in order to highlight the synoptic forcing. For $\Delta Z_{800\,hPa}$, this filtered signal exceeded 15 m (grey dash-dotted line in Fig. 3b) during the strongest winds in the GRV. The peak in $\Delta Z$ at 800 hPa in Fig. 3a on 14 January was therefore composed ...'
Concerning the southwest case, the manuscript (L253) reads now: '... which is similar to the northeast gap flow. The synoptic forcing below pass height, represented by the 24-hour running mean of $\Delta Z_{800\,hPa}$ in Fig. 7b, showed a similar behavior as for the northeast case and exceeded 15 m. However, even before the investigated case the 24-hour average of $\Delta Z_{800\,hPa}$ was close to 15 m and, hence, caused southwesterly winds in the GRV (Fig. 7d). Similar to the northeast case, the magnitude of the 24-hour running mean of $\Delta Z$ was much larger at 800 hPa than at 700 hPa (Fig. 7b), indicating that the large-scale forcing was

strongest below crest height, i.e. within the GRV.'
We added a dash-dotted line in Fig. 3 and 7 for $\Delta Z = 15$ m to emphasis the possible threshold for the synoptic forcing.

- Response to RC on 3.3. (7) of reviewer 2.
I agree with the authors' explanation. However, this information does not seem to be present in the manuscript. I think this information needs to be included in the manuscript as well for the benefit of readers who are not familiar with this area.

We changed that. The manuscript (L151) reads now: '...the Turkana Jet (Nicholson, 2016; Munday et al., 2022) brought potentially colder air into the Turkana Channel and the region around Lake Abaya (Fig. 2b). This air originated from the southern Indian Ocean and was deflected by the massive of Mt. Kenya towards the northeast, resulting in the southwesterly winds on Fig. 2b. Below 800 hPa, a branch of this southwesterly air stream was deflected by the Turkana Jet into the Turkana Channel (not shown). The different air masses...'

-Response to RC on 3.3. (16) of reviewer 2.
I understand. However, while the specific numbers of the distances are of course important, it is necessary to write "between where" in the manuscript so that "the readers can understand".

Thanks, now we got the meaning of the initial comment - it was not about the specific distance but rather about geographic landmarks. We changed it and the sentence (L334) reads now: 'The difference in geopotential height at 800 hPa (i.e., near pass height) over a horizontal distance of more than 700 km between the Afar Triangle and the Turkana Channel exceeded in both cases 30 m for the daily maximum and 15 m for the 24-hour average.'

Response to RC on 3.3. (19)-(2) of reviewer 2 (Just a comment, no need to reflect the manuscript).
I am sorry that I could not accurately inform you of my intentions. In the evening (14 UTC, Fig. 9a), gap winds develop near AG (between x=550 and x = 650 km). On the other hand, at night (19UTC, Fig. 9c), the gap winds near AG are weak and gap winds develop between x=100 and x=500 km. In other words, in the evening (14UTC), the location of the gap winds is limited to the vicinity of AG due to the thermally localized circulation (pressure gradient that drives the plateau and basin winds), and at night (19UTC), the gap winds occur over the entire valley because the thermally localized circulation decays. Is it correct? The authors think that this is probably future studies. I am looking forward to it.

Yes, it is conceivable that counteracting basin winds from the northeast into the GRV north of the pass could have prevented an earlier breakthrough or at least weakened the southwesterly gap winds in the northern part of the valley during the day. However, we did not find strong evidence for basin winds. Concerning the weak gap winds around AG in the night: this is most likely due to the flow blocking below pass height indicated by the isentropes intersecting the slope between PA and AG in Fig 9c.

**3. RESPONDS TO CO-EDITOR'S COMMENTS**

Comments to the author: Dear C. Weiss et al.,

Thank you for submitting your revised manuscript and detailed response letter. You may see that both reviewers are positive, however, the second referee suggests further minor revisions for better clarity for the readers.

Please note that both reviewers raised concern regarding deltaZ at 800 hPa as a diagnostic for the gap winds, especially given that the diagnostic peaks also prior to the onset of the gap winds, e.g., during 12 January. The addition of the smoothed values as panel 3b in the revised manuscript partly addresses this issue, and indeed shows that a clearer peak emerges on 14 January. However, accompanying text is needed in the manuscript as well, as other readers may have the same concern. I agree with the second referee that a more elaborate discussion of the atmospheric structure during the days prior to onset is necessary to complete this description.

Furthermore, it is the aim of the study to reveal whether the flow is dynamically- or thermally-driven. However, using the simulation set it is not possible to disentangle the components, as noted by the second referee as well. I therefore suggest that you rephrase lines 80-82 along the lines of the conclusions of this study.

Looking forward to receiving your revised manuscript and response, Shira

Thank you for your feedback. We hope that our changes sufficiently address your remarks.

Regarding $\Delta Z$: We elaborated the discussion on $\Delta Z$ and its 24-hour running mean for both cases. For the changes see Sec. 2.2 - 'Response to RC on Major comment (2) of reviewer 2' in this document.

Regarding the goals of the study: We rephrased the conclusive paragraph in the introduction. The manuscript (L77) reads now: '...the more specific goals of this study are to investigate the physical mechanisms of the two cases and their daytime and seasonal dependency. The paper is organized...'

Additionally, we added following paragraph at the end of the discussion (L400):

'Finally, it is noteworthy that the underestimation of the gap flow in the GRV in ERA5 agrees with the underestimation of the Turkana Jet recently found by Munday et al. (2022) based on radiosonde observations. Similar to the gap flow in the GRV, the Turkana Jet does not only exhibit a strong diurnal cycle due to the thermal forcing (Munday et al., 2022) but is also affected by synoptic-scale pressure gradients associated with low-level ridging along the East African coast (Vizy and Cook, 2019).'

This does not only show that another mesoscale wind phenomenon in the same region is strongly dependent on the time of the day but that ERA5 underestimates that phenomenon as well.

**References**

Munday, C., Engelstaedter, S., Ouma, G., Ogutu, G., Olago, D., Ong'ech, D., Lees, T., Wanguba, B., Nkatha, R., Ogalo, C., Gàlgalo, R. A., Dokata, A. J., Kirui, E., Hope, R., and Washington, R.: Observations of the Turkana Jet and the East African dry tropics: the RIFTJet field campaign, Bulletin of the American Meteorological Society, https://doi.org/10.1175/BAMS-D-21-0214.1, 2022.

Nicholson, S.: The Turkana low-level jet: Mean climatology and association with regional aridity, International Journal of Climatology, 36, 2598–2614, https://doi.org/10.1002/joc.4515, 2016.

Vizy, E. K. and Cook, K. H.: Observed relationship between the Turkana low-level jet and boreal summer convection, Climate Dynamics, 53, 4037–4058, https://doi.org/https://doi.org/10.1007/s00382-019-04769-2, 2019.